# Complement and CD4[+] T cells drive context-specific corneal sensory neuropathy

Derek J Royer[1,2†]*, Jose Echegaray-Mendez[1], Liwen Lin[1], Grzegorz B Gmyrek[2], Rose Mathew[1], Daniel R Saban[1,3], Victor L Perez[1], Daniel JJ Carr[2,4]

[1]Department of Ophthalmology, Duke University Medical Center, Durham, United States; [2]Department of Ophthalmology, University of Oklahoma Health Sciences Center, Oklahoma City, United States; [3]Department of Immunology, Duke University Medical Center, Durham, United States; [4]Department of Microbiology and Immunology, University of Oklahoma Health Sciences Center, Oklahoma City, United States

*For correspondence:
Derek.Royer@Duke.edu

Present address: [†]Department of Ophthalmology, Duke Eye Center, Durham, United States

Competing interests: The authors declare that no competing interests exist.

**Abstract** Whether complement dysregulation directly contributes to the pathogenesis of peripheral nervous system diseases, including sensory neuropathies, is unclear. We addressed this important question in a mouse model of ocular HSV-1 infection, where sensory nerve damage is a common clinical problem. Through genetic and pharmacologic targeting, we uncovered a central role for C3 in sensory nerve damage at the morphological and functional levels. Interestingly, CD4 T cells were central in facilitating this complement-mediated damage. This same C3/CD4 T cell axis triggered corneal sensory nerve damage in a mouse model of ocular graft-versus-host disease (GVHD). However, this was not the case in a T-dependent allergic eye disease (AED) model, suggesting that this inflammatory neuroimmune pathology is specific to certain disease etiologies. Collectively, these findings uncover a central role for complement in CD4 T cell-dependent corneal nerve damage in multiple disease settings and indicate the possibility for complement-targeted therapeutics to mitigate sensory neuropathies.
DOI: https://doi.org/10.7554/eLife.48378.001

## Introduction

Dysregulated complement activation is increasingly recognized as a significant pathological event in a variety of neurodegenerative and neuroinflammatory diseases of the central nervous system (*Tenner et al., 2018*). These include conditions from Alzheimer's disease and age-related macular degeneration (AMD) to amyotrophic lateral sclerosis and multiple sclerosis (*Hong et al., 2016*; *Knickelbein et al., 2015*; *Lee et al., 2018*; *Loveless et al., 2018*). Complement may orchestrate peripheral nervous system (PNS) disorders as well. For instance, nociceptive hypersensitivities, sensory neuropathies, and Guillain-Barré syndrome have been linked to aberrant complement activation (*Rosoklija et al., 2000*; *Ramaglia et al., 2008*; *Jang et al., 2010*; *Fritzinger and Benjamin, 2016*; *Xu et al., 2018*; *Susuki et al., 2007*). However, the mechanistic role of complement is not well understood in neuroinflammatory pathologies impacting the PNS. Moreover, the pathophysiological trigger of sensory fiber retraction which contributes to sensation loss and in some instances chronic pain is unclear (*Stepp et al., 2017*). Available treatments for peripheral neuropathies primarily focus on clinical symptoms without addressing the underlying pathomechanisms (*Colloca et al., 2017*). Accordingly, elucidation of relevant pathomechanisms underlying sensory neuropathies may translate into efficacious mechanism-based therapeutics.

**eLife digest** Most people have likely experienced the discomfort of an eyelash falling onto the surface of their eye. Or that gritty sensation when dust blows into the eye and irritates the surface. These sensations are warnings from sensory nerves in the cornea, the transparent tissue that covers the iris and pupil. Corneal nerves help regulate blinking, and control production of the tear fluid that protects and lubricates the eye.

But if the cornea suffers damage or infection, it can become inflamed. Long-lasting inflammation can damage the corneal nerves, leading to pain and vision loss. If scientists can identify how this happens, they may ultimately be able to prevent it. To this end, Royer et al. have used mice to study three causes of hard-to-treat corneal inflammation. The first is infection with herpes simplex virus (HSV-1), which also causes cold sores. The second is eye allergy, where the immune system overreacts to substances like pollen or pet dander. And the third is graft-versus-host disease (GVHD), an immune disorder that can affect people who receive a bone marrow transplant.

Royer et al. showed that HSV-1 infection and GVHD – but not allergies – made the mouse cornea less sensitive to touch. Consistent with this, microscopy revealed damage to corneal nerves in the mice with HSV-1 infection and those with GVHD. Further experiments showed that immune cells called CD4 T cells and a protein called complement C3 were contributing to this nerve damage. Treating the mice with an experimental drug derived from cobra venom protected the cornea from the harmful effects of inflammation. It did so by blocking activation of complement C3 at the eye surface.

Identifying factors such as complement C3 that are responsible for corneal nerve damage is an important first step in helping patients with inflammatory eye diseases. Many drugs that target the complement pathway are currently under development. Some of these drugs could potentially be adapted for delivery as eye drops. But first, experiments must test whether complement also contributes to corneal nerve damage in humans. If it does, work can then begin on testing these drugs for safety and efficacy in patients.

DOI: https://doi.org/10.7554/eLife.48378.002

The complement cascade is comprised of dozens of soluble and membrane-bound factors essential for host defense and efficient clearance of cellular debris. However, proper regulation of complement activation is necessary to balance immune responses and prevent collateral tissue damage (*Ricklin et al., 2016*). The complement cascade pivots upon activation of complement component 3 (C3) for its downstream proinflammatory effects. The pathogenic contributions of complement in systemic disease have been reviewed extensively in recent years (*Dobó et al., 2018*; *Hajishengallis et al., 2017*; *McGeer et al., 2017*; *Ricklin et al., 2016*). Nonetheless, emerging concepts centering on complement effector synthesis within inflamed tissue microenvironments have ushered in a 'renaissance' of novel complement-targeted drug development strategies (*Ricklin et al., 2018*; *Tomlinson and Thurman, 2018*). Therefore, elucidating whether the complement cascade is a viable therapeutic target in inflammation-associated peripheral nerve impairment remains medically important.

The eye has been labeled a 'complement dysregulation hotspot' due to complement's contributions to many ophthalmic diseases, but complement activation is restricted in the healthy cornea (*Clark and Bishop, 2018*). As observed in AMD, subtle inflammatory reactions in the eye can mediate significant visual morbidity. Accordingly, ocular inflammation is tightly regulated by various anatomic, physiologic, and immunologic mechanisms in order to maintain visual acuity (*Amouzegar et al., 2016*; *Streilein, 2003*; *Taylor et al., 2018*). These homeostatic mechanisms, collectively dubbed 'ocular immune privilege,' help preserve the ocular surface and enable the cornea to properly focus incoming light onto the retina. Furthermore, corneal nerves are increasingly recognized as central regulators of immune privilege at the ocular surface (*Guzmán et al., 2018*; *Neelam et al., 2018*; *Paunicka et al., 2015*). Consequently, inflammatory events that damage corneal nerves can have insidious consequences in terms of ocular surface health and transparency (*Müller et al., 2003*; *Shaheen et al., 2014*; *Stepp et al., 2017*). One such pathway that can rapidly initiate inflammation following activation is the complement cascade. Corneal nerves originate

predominantly from the ophthalmic branch of the trigeminal ganglion (TG), and these peripheral nerves provide the cornea with the highest density of sensory fibers in the human body. While the cornea and peripheral nerves constitutively synthesize complement proteins (*Bora et al., 2008*; *de Jonge et al., 2004*), the role of the complement cascade in corneal nerve damage has not been explored. Nonetheless, the complement pathway is poised for vigorous activation in the cornea in response to noxious or inflammatory stimuli (*Bora et al., 2008*). Moreover, C3 activation has been reported in the cornea soon after infection with herpes simplex virus type 1 (HSV-1), which is a common cause of corneal sensory nerve damage in patients (*Royer et al., 2017*; *Sacchetti and Lambiase, 2014*). Given the cornea's high density of sensory nerves, it is particularly amenable for investigating the mechanistic role of complement in peripheral nerve damage. To this end, we evaluated corneal nerve integrity and mechano-sensory function using a murine model of ocular HSV-1 infection to test the hypothesis that local complement activation and T cell engagement coordinate corneal nerve damage.

The pathobiology underlying non-penetrating corneal nerve damage is not well understood, although inflammation is generally recognized as an important feature (*Neelam et al., 2018*; *Shaheen et al., 2014*; *Cruzat et al., 2015*; *Chucair-Elliott et al., 2016*; *Chucair-Elliott et al., 2017b*; *Chucair-Elliott et al., 2017a*). While HSV-1 is a clinically prominent cause of corneal nerve damage and sensation loss, a variety of pathogenic microbes impair the corneal nerve architecture upon ocular infection (*Cruzat et al., 2015*). This observation adds to evidence that the neurotropism of HSV-1 is not directly responsible for nerve damage (*Chucair-Elliott et al., 2016*). In addition to sensation loss evoked by HSV-1, corneal nerve alterations in other contexts such as dry eye disease are associated with a broad array of neuropathic clinical symptoms including dryness, itch, and pain (*Andersen et al., 2017*). To further qualify the possible role of complement in corneal nerve damage independent of infection, we evaluated corneal mechanosensory function in two noninfectious T cell-dependent ocular surface inflammatory diseases. For this purpose, we utilized established murine models of allergic eye disease (AED) and ocular graft-versus-host disease (GVHD) (*Herretes et al., 2015*; *Lee et al., 2015*). Our rationale for this is that complement has been implicated in the etiology of systemic GVHD and allergic inflammation (*Gour et al., 2018*; *Kwan et al., 2012*; *Ma et al., 2014*; *Nguyen et al., 2018*; *Zhang and Köhl, 2010*). However, a neuroinflammatory role of complement has not been described for either disease within the eye.

The translational relevance of this study is underscored by in vivo confocal microscopy data from multiple clinical studies showing architectural changes in the corneal nerves of patients with herpetic keratitis, chronic ocular allergy, and ocular GVHD (*Hamrah et al., 2010*; *Müller et al., 2015*; *Moein et al., 2018*; *Hu et al., 2008*; *Le et al., 2011*; *Leonardi et al., 2012*; *Tepelus et al., 2017*; *He et al., 2017a*). Elucidating the pathomechanisms underlying corneal nerve damage may enable development of more effective therapeutics to mitigate progression of such ocular surface inflammatory diseases. Each of the animal models utilized herein mimic clinically important, chronic ocular surface morbidities. Herpes simplex virus type 1 (HSV-1) is a common cause of neurotropic keratitis and remains a leading cause of infectious corneal blindness (*Sacchetti and Lambiase, 2014*; *Farooq and Shukla, 2012*). The incidence of ocular allergy exceeds twenty percent of the population with varying degrees of neurogenic ocular surface discomfort that can severely diminish quality of life (*Craig et al., 2017*; *Patel et al., 2017*; *Saban et al., 2013*). Finally, GVHD is the greatest cause of non-relapse morbidity following hematopoietic stem cell transplantation (HSCT) used to treat life-threatening malignancies and immunologic diseases (*MacDonald et al., 2017*). A majority of patients with chronic presentations of GVHD suffer from ocular surface involvement (*Shikari et al., 2013*).

## Results

### C3 facilitates corneal sensation loss in herpetic keratitis

Adaptive immunity has been shown to *prevent* recovery of corneal sensory function following ocular HSV-1 infection (*Yun et al., 2014*), but the initial pathophysiological triggers of denervation and sensation loss are not definitively characterized. Ocular HSV-1 infection provokes corneal denervation and sensation loss between days 5 to 8 post-infection (p.i.) in immunologically naive C57BL/6 mice (*Chucair-Elliott et al., 2015*). This tempo appears to be synchronized with the host transition from

innate to adaptive immunity following infection. Indeed, corneal nerve fiber retraction was evident upon T cell infiltration in the cornea (*Figure 1A*). Accordingly, we hypothesized that local complement activation and T cell engagement coordinate corneal nerve damage during HSV-1 infection.

To address our hypothesis, corneal sensation and pathogen burden were evaluated in C57BL/6 wildtype (WT) and complement C3-deficient (C3$^{-/-}$) mice following ocular HSV-1 infection. Progressive loss of corneal mechano-sensitivity was evident in WT mice by days 5 to 8 p.i., but corneal sensation was conserved in the C3$^{-/-}$ cohort (*Figure 1B*). The same trend revealing preservation of corneal sensation in C3$^{-/-}$ animals was also observed with an increased HSV-1 challenge inoculum (*Figure 1—figure supplement 1*). Viral titers were measured at time points before and after the onset of corneal sensation loss to determine whether the divergence in sensation stemmed from a

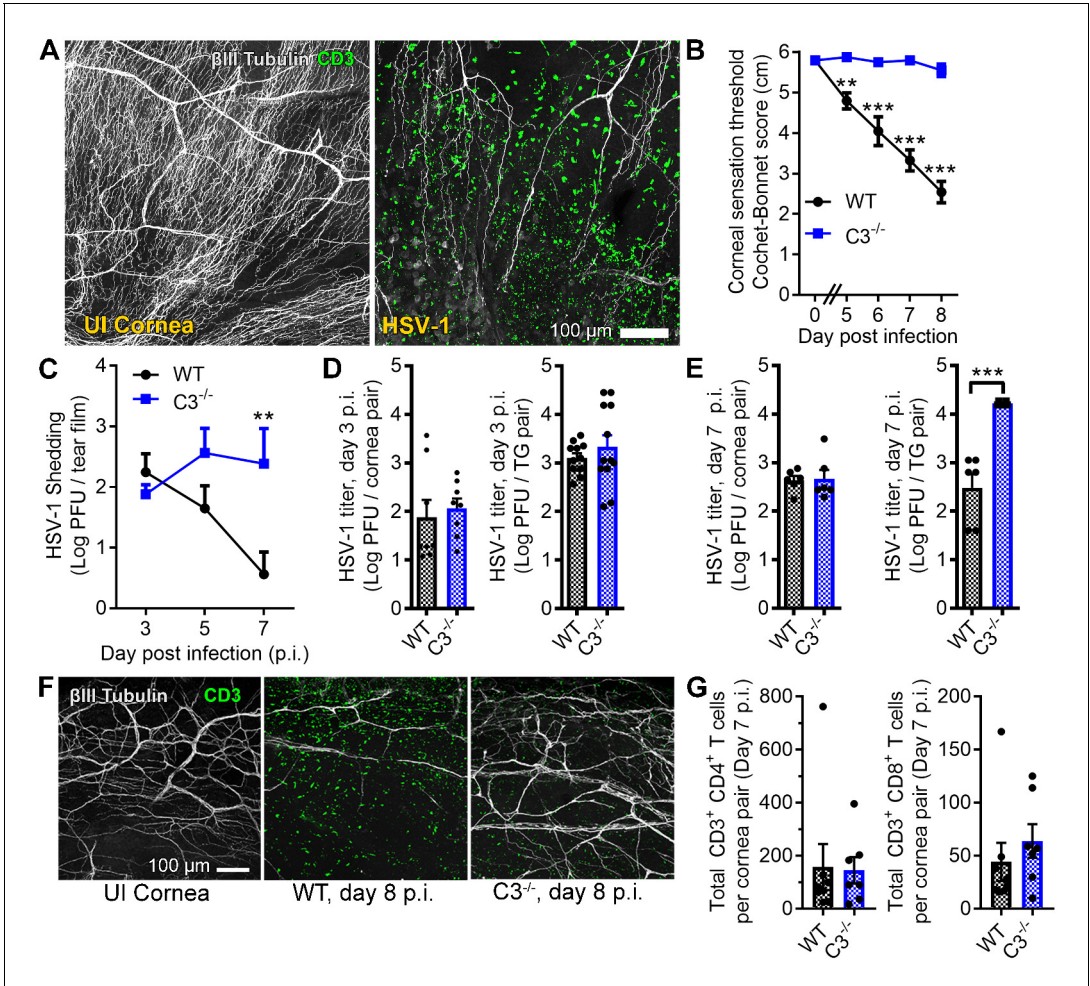

**Figure 1.** Complement C3 contributes to corneal denervation. (**A**) Representative confocal images of cornea flat-mounts showing corneal nerves (βIII Tubulin, white) and T cells (CD3, green) in healthy uninfected (UI) and HSV-1-infected corneas 8 days post infection (p.i.). (**B**) Corneal mechanosensory function in WT and C3$^{-/-}$ mice following ocular HSV-1 infection (*n* = 6–8 mice/group; three independent experiments). (**C**) Viral titers shed in the tear film of WT and C3$^{-/-}$ mice at the indicated times p.i. (*n* = 5 mice per group; two independent experiments). Viral titers in the corneas and trigeminal ganglia (TG) of WT and C3$^{-/-}$ mice are shown at days 3 and 7 p.i. in (**D**) and (**E**), respectively. (**F**) Representative confocal images of stromal nerve fibers and T cells in healthy and HSV-infected corneas of WT and C3$^{-/-}$ mice as in (**A**). (**G**) Flow cytometry-based quantification of CD4$^+$ and CD8$^+$ T cells in HSV-infected corneas at day 7 p.i. (*n* = 7–8 mice/group; four independent experiments). Statistical differences were determined using two-way ANOVAs with Bonferroni posttests (**B, C**) or Student's T tests (**D, E, G**).

DOI: https://doi.org/10.7554/eLife.48378.003

The following figure supplement is available for figure 1:

**Figure supplement 1.** C3$^{-/-}$ mice retain corneal sensation through viral latency.

DOI: https://doi.org/10.7554/eLife.48378.004

differential susceptibility to infection. On day 3 p.i., the HSV-1 burden was similar in the tear film, corneas, and TG of WT and C3⁻/⁻ animals (*Figure 1C–D*). By day 7 p.i., HSV-1 titers were elevated in the tear film and TG of C3⁻/⁻ mice relative to WT (*Figure 1C,E*). Corneal buttons from WT mice exhibited extensive denervation with CD3⁺ T cell infiltration at day 8 p.i. (*Figure 1F*). However, T cell infiltration was observed without widespread denervation in C3⁻/⁻ corneas (*Figure 1F*). The number of cornea-infiltrating CD4⁺ or CD8⁺ T cells were similar among WT and C3⁻/⁻ mice (*Figure 1G*). Together, these data indicate that corneal sensation loss in herpetic keratitis involves a C3-dependent inflammatory process independent of viral burden.

## Corneal sensation loss during HSV-1 infection requires T cell coordination

Although T cells successfully extravasate into the corneas of WT and C3⁻/⁻ mice (*Figure 1F,G*), C3-deficiency can have broad impacts on T cell priming, clonal expansion, and recruitment (*Clarke and Tenner, 2014*; *West et al., 2018*). To investigate this possibility, corneas from HSV-1-infected WT and C3⁻/⁻ mice were evaluated for chemokines associated with T cell recruitment, and the eye-draining mandibular lymph nodes (MLN) were harvested to evaluate T cell responses. Chemokines associated with T cell recruitment were elevated in corneas from both WT and C3⁻/⁻ mice at day 5 p.i.—a time consistent with onset of sensation loss (*Figure 2A*). Moreover, CD4⁺ and CD8⁺ T cell expansion was comparable within the eye-draining mandibular lymph nodes (MLN) from WT and C3⁻/⁻ mice (*Figure 2B*). Similarly, T cells harvested from WT and C3⁻/⁻ mice at day 8 p.i. responded equally to in

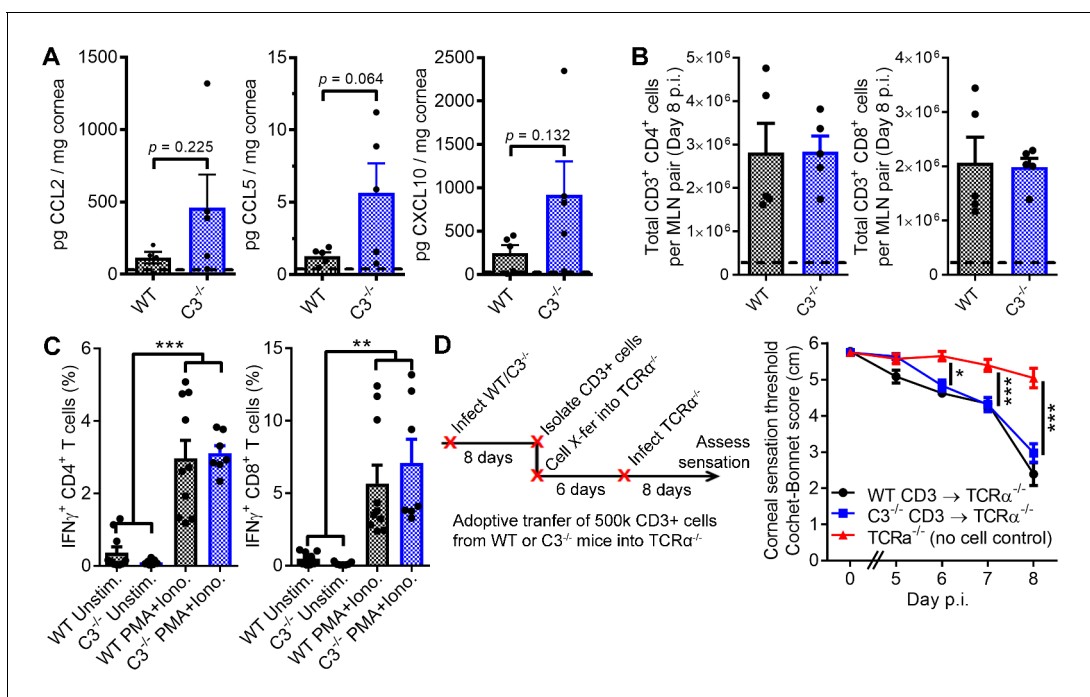

**Figure 2.** T cells facilitate corneal sensation loss. (**A**) Chemokine concentrations in HSV-1-infected corneas from WT and C3⁻/⁻ mice at day five post infection (p.i.). (**B**) T cell expansion in the eye-draining mandibular lymph nodes of WT and C3⁻/⁻ mice. For panels (**A**) and (**B**), dashed lines reflect the average value for uninfected WT controls (n = 5 mice per group; two independent experiments; Student's T). (**C**) IFNγ expression following stimulation with PMA and ionomycin using T cells harvested from WT or C3⁻/⁻ mice at day eight post infection (n = 7 unstimulated and 10 activated replicates from three independent experiments; one-way ANOVA, Bonferroni). (**D**) Adoptive transfer schematic and corneal sensation measurements in TCRα⁻/⁻ mice following reconstitution with purified splenic T cells from HSV-infected WT and C3⁻/⁻ mice (n = 5–9 TCRα⁻/⁻ mice/group; three independent experiments; two-way ANOVA, Bonferroni).

DOI: https://doi.org/10.7554/eLife.48378.005

The following figure supplement is available for figure 2:

**Figure supplement 1.** T cell engraftment in recipient TCRα⁻/⁻ mice.

DOI: https://doi.org/10.7554/eLife.48378.006

vitro stimulation in terms of IFNγ production (*Figure 2C*). Collectively, these data show that C3$^{-/-}$ T cells exhibit normal expansion and recruitment to the cornea during ocular HSV-1 infection.

In support of our hypothesis that C3 and T cells jointly coordinate corneal sensation loss, frank corneal sensation loss was not observed following ocular HSV-1 infection in C3-sufficient, alpha-beta T cell receptor-deficient mice (TCRα$^{-/-}$) which congenitally lack classical CD4$^+$ and CD8$^+$ T cells (*Figure 2D*). To investigate potential functional defects of C3$^{-/-}$ T cells in an in vivo context, CD3$^+$ T cells were harvested from HSV-infected WT or C3$^{-/-}$ mice at day 8 p.i. and adoptively transferred into TCRα$^{-/-}$ mice. Six days following cell transfer, TCRα$^{-/-}$ recipients were infected with HSV-1 and corneal sensation monitored longitudinally. Adoptive transfer of CD3$^+$ T cells from either WT or C3$^{-/-}$ donors evoked progressive corneal sensation loss in TCRα$^{-/-}$ recipients by day 8 p.i. (*Figure 2D*). These findings further corroborate that C3$^{-/-}$ T cells remain functional in vivo. Analysis of corneal sensation was not feasible beyond 8 days p.i., as the TCRα$^{-/-}$ mice succumbed to herpetic encephalitis. However, engraftment of donor cells was confirmed by analysis of T cells in the eye-draining MLN of recipient mice by flow cytometry at day 8 p.i. The baseline lymphocyte counts in MLN from TCRα$^{-/-}$ controls likely reflect expansion of non-classical T cell populations (*Viney et al., 1994*). An increase in total cell number was observed for CD4$^+$ but not CD8$^+$ T cells upon adoptive transfer of purified CD3$^+$ T cells into TCRα$^{-/-}$ recipient mice (*Figure 2—figure supplement 1A*). Consistent with this observation, our data show that transfer of CD3$^+$ T cells from HSV-infected mice have no appreciable impact on HSV-1 titers in the TG of TCRα$^{-/-}$ recipients by day 8 p.i. (*Figure 2—figure supplement 1B*). Notably, bulk transfer of HSV-specific CD8$^+$ T cells has been shown to reduce viral burden in the peripheral nervous system of C57BL/6 mice during acute HSV-1 infection (*Conrady et al., 2009*; *Royer et al., 2016*). Our collective findings reveal that C3 and T cells are individually necessary and interdependent in the pathogenesis of HSV-1-associated corneal sensation loss. Whether by direct or indirect mechanisms, these data support a paradigm in which C3 activation and T cell engagement coordinate corneal nerve damage in herpetic keratitis.

## Antigen-specific CD4$^+$ T cells facilitate corneal sensation loss in HSV-1 keratitis

Our data show that T cells contribute to corneal sensation loss during acute HSV-1 infection, yet whether this pathology is dependent upon CD4$^+$ or CD8$^+$ T cells remained unclear. To better discern the contributions of each subset, CD4$^+$ and CD8$^+$ T cells were harvested from infected WT mice, transferred into separate groups of TCRα$^{-/-}$ mice, and recipient mice subsequently infected with HSV-1. Corneal sensation loss during acute HSV-1 infection was only observed upon reconstitution with CD4$^+$ T cells (*Figure 3A*). Donor cell engraftment was confirmed by flow cytometry, although a statistically significant increase in cell number within the eye-draining MLN was only observed upon transfer of CD4$^+$ T cells (*Figure 3—figure supplement 1*). The requirement for antigen specificity was subsequently evaluated during acute HSV-1 infection by monitoring corneal sensation in WT and OT-II transgenic mice that generate ovalbumin (OVA)-specific CD4 T cells. Transgenic OT-II mice did not exhibit corneal sensation loss following infection despite evidence of bystander CD4$^+$ T cell activation and recruitment into the cornea (*Figure 3B–D*). Adoptive transfer of CD4$^+$ T cells from HSV-infected WT and OT-II mice into TCRα$^{-/-}$ recipients corroborated these findings (*Figure 3—figure supplement 2*). Abrasion of the corneal epithelium, which is required to mediate infection in this model, induces local chemokine expression capable of recruiting T cells to the cornea even in the absence of viral infection (*Liu et al., 2012*). Nonetheless, active corneal HSV-1 infection was necessary to provoke corneal sensation loss, as TCRα$^{-/-}$ mice reconstituted with CD4$^+$ T cells harvested from HSV-infected WT mice did not elicit auto/allo-antigen-associated sensation loss within two weeks of corneal scratch injury (i.e. mock infection) (*Figure 3—figure supplement 3*). Collectively, our data show that complement C3 and antigen-specific CD4$^+$ T cells are simultaneously necessary to drive corneal sensation loss during HSV-1 infection. These results strongly portend that a coordinated inflammatory axis exists involving C3 and antigen-specific CD4$^+$ T cells, and that it is responsible for sensory neuropathy in herpetic keratitis.

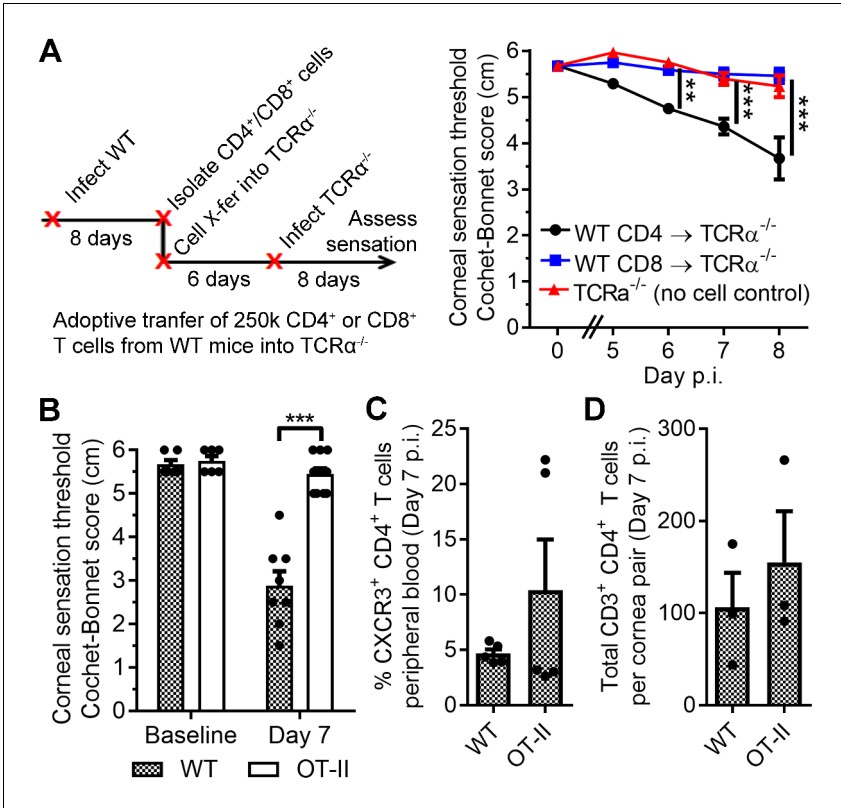

**Figure 3.** Antigen-specific CD4 T cells drive corneal sensation loss. (**A**) Adoptive transfer schematic and corneal sensation measurements in TCRα$^{-/-}$ mice following reconstitution with purified splenic CD4 or CD8 T cells from HSV-infected WT mice ($n$ = 6–7 TCRα$^{-/-}$ mice/group; two independent experiments; two-way ANOVA, Bonferroni). (**B**) Corneal sensation measurements at baseline and day seven post infection (p.i.) in WT and OT-II mice following ocular HSV-1 infection ($n$ = 4–6 mice/group; three independent experiments). (**C**) Percentage of CXCR3-expressing CD4 T cells in peripheral blood from WT and OT-II mice at day 7 p.i.; ($n$ = 5 mice/group; two independent experiments). (**D**) Verification of CD4 T cell infiltration into corneas of WT and OT-II mice at day 7 p.i. ($n$ = 3 mice/group; two independent experiments). Data in panels B-D were analyzed using Student's T tests.
DOI: https://doi.org/10.7554/eLife.48378.007

The following figure supplements are available for figure 3:

**Figure supplement 1.** Engraftment of donor T cells into TCRα$^{-/-}$ mice.
DOI: https://doi.org/10.7554/eLife.48378.008

**Figure supplement 2.** Antigen-specific donor CD4 T cells drive sensation loss during HSV-1 keratitis.
DOI: https://doi.org/10.7554/eLife.48378.009

**Figure supplement 3.** Active corneal HSV-1 infection is required to drive sensation loss in herpetic keratitis.
DOI: https://doi.org/10.7554/eLife.48378.010

## Nonhematopoietic cornea-resident cells and CSF1R$^+$ leukocytes augment local C3 synthesis during herpetic keratitis

Complement-mediated tissue pathology can arise from complement activators synthesized by the liver (systemic complement) or in tissue microenvironments (local complement). While local complement activation and regulation have been reported in the cornea in health and disease (*Bora et al., 1993*; *Clark and Bishop, 2018*; *Mondino and Brady, 1981*; *Verhagen et al., 1992*), the expression profile and cellular sources of various complement components have not been investigated in the context of ocular HSV-1 infection. To this end, a modest array of complement component transcripts was evaluated by semiquantitative real-time PCR on corneal buttons from healthy and HSV-infected mice at day 2 and 7 p.i. Upregulation of complement effectors and anaphylatoxin receptors were noted including *C3*, *C5*, *C3ar1*, and *C5ar1* (*Figure 4A,B*). Constitutive expression of complement receptor 2 (CD21) has been reported in the corneal epithelium (*Levine et al., 1990*), but variances in

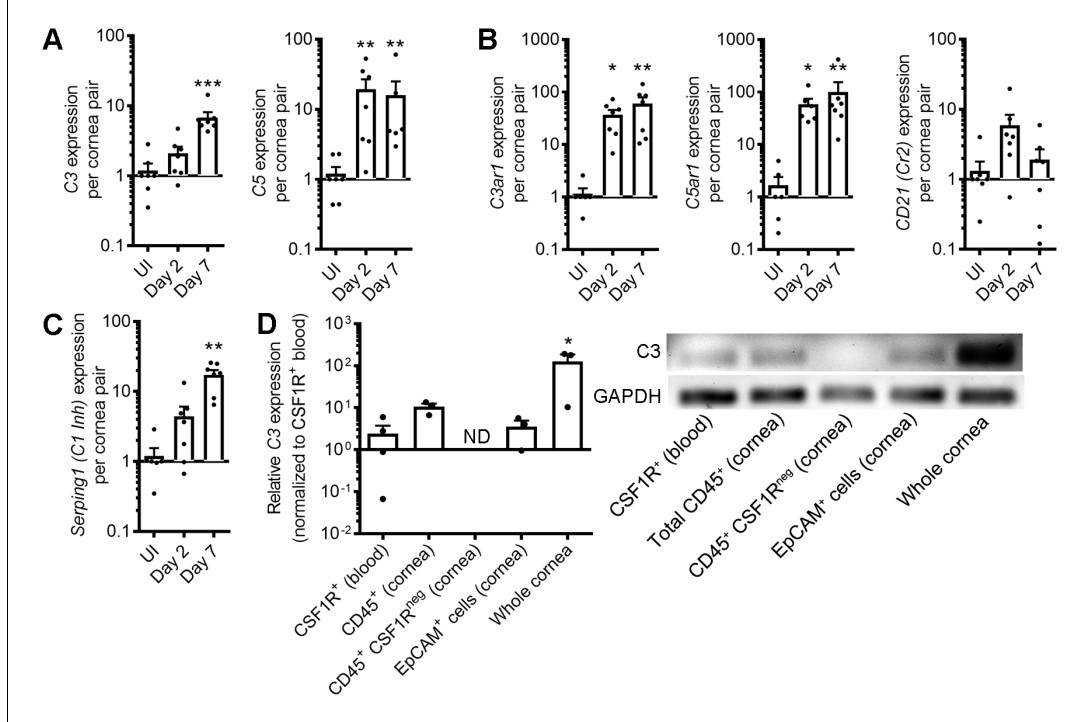

**Figure 4.** Corneal HSV-1 infection enhances local complement synthesis. Gene expression of complement effectors (**A**), receptors (**B**), and regulators (**C**) upregulated in the corneas of B6 mice during acute HSV-1 infection (n = 7 WT mice/group; two independent experiments; Kruskal-Wallis, Dunn's multiple comparisons test). (**D**) Relative *C3* expression among selected cornea-resident and infiltrating cell subsets at day three post-infection (n = 3–4 pooled samples from two mice each for cell subsets or three independent cornea pairs; two independent experiments; one-way ANOVA, Bonferroni; ND, not detected/amplification cycle >35). Final PCR products were resolved on an agarose gel to verify amplification. Data are relative to GAPDH expression and normalized to uninfected control samples for panels A-C or to purified CSF1R-expressing peripheral blood monocytes/macrophages in panel D.

DOI: https://doi.org/10.7554/eLife.48378.011

its expression during HSV-1 infection were not statistically significant across the time points evaluated (*Figure 4B*). Likewise, no differences in the local expression of various complement regulators were observed in HSV-infected corneas aside from the C1-inhibitor *SerpinG1* (*Table 1*, *Figure 4C*). Collectively, this expression profile favors local complement activation, as effectors are upregulated without proportional enhancement of pathway regulatory components.

**Table 1.** Complement regulatory factor expression in the cornea during HSV-1 infection

| Gene | Uninfected | Day 2 p.i. | Day 7 p.i. |
|---|---|---|---|
| *C4bp* | ND | ND | ND |
| *Crry (CD46 homolog)* | 1.193 ± 0.338 | 0.817 ± 0.172 | 1.137 ± 0.331 |
| *CD55 (DAF)* | 1.087 ± 0.201 | 1.035 ± 0.199 | 0.791 ± 0.251 |
| *CD59a (MIRL)* | 1.078 ± 0.192 | 0.611 ± 0.123 | 0.728 ± 0.191 |
| *Cfi* | ND | ND | ND |
| *Cr1l* | 1.026 ± 0.096 | 0.7442 ± 0.222 | 0.889 ± 0.133 |

Data are expressed as mean ± SEM; *n* = 7 samples/group. Gene expression is standardized to internal *GAPDH* expression and relative to uninfected control tissue. Abbreviations: C4bp, C4 binding protein; CD, cluster of differentiation; Cfi, complement factor I, Cr1l, complement C3b/C4b receptor-1-like; DAF, decay accelerating factor; GAPDH, glyceraldehyde 3-phosphate dehydrogenase; MIRL, membrane inhibitor of reactive lysis; ND, not detected (amplification cycle >35); p.i., post-infection.

DOI: https://doi.org/10.7554/eLife.48378.012

As C3 is the central component of the complement activation pathway, the cellular sources of complement C3 were evaluated in corneas infected with HSV-1. Monocytes and tissue macrophages are known sources of C3 (*Einstein et al., 1977*; *Lubbers et al., 2017*; *Morgan and Gasque, 1997*; *Verschoor et al., 2001*), thus CSF1R⁺ cells from the peripheral blood of transgenic CSF1R-GFP mice (MAFIA) were utilized as the relative standard for *C3* transcript expression. Expression of *C3* was noted in CD45⁺ leukocytes isolated from infected corneas, yet detection of *C3* expression was lost when CSF1R⁺ cells were removed from the total CD45⁺ pool. In addition, isolated EpCAM⁺ corneal epithelial cells expressed *C3* at levels comparable to blood monocytes. However, the *C3* expression level was greater in whole-cornea preparations than among individual cell fractions (*Figure 4D*). Taken together, our data show that both tissue-resident non-hematopoietic cells and resident/infiltrating CSF1R⁺ leukocytes contribute to local *C3* expression in the cornea during acute HSV-1 infection.

## Localized pharmacologic C3 depletion preserves corneal sensation in HSV-1 keratitis

Ocular HSV-1 infection amplifies local complement gene expression, yet whether local control of complement activation can be harnessed to prevent corneal nerve damage has not been explored. From a clinical perspective, modulating complement activation at the ocular surface may be a viable therapeutic option. As a proof of concept, daily ocular cobra venom factor (CVF) treatment was explored as a putative method to deplete C3 and preserve corneal sensation during acute HSV-1 infection. Vehicle (PBS)-treated mice exhibited corneal sensation loss following HSV-1 infection, but CVF treatment preserved corneal sensation (*Figure 5A*). Protein levels of C3 remained near baseline in the cornea following CVF treatment during HSV-1 infection, yet vehicle-treated animals exhibited a 300–600% increase in C3 protein levels in the cornea at days 3 and 7 p.i. (*Figure 5B*). Ocular CVF treatment did not significantly impact systemic serum C3 concentrations throughout the study (*Figure 5C*).

Ocular CVF treatment limited corneal edema during HSV-1 infection relative to the vehicle control (*Figure 5D*). Consistent with this observation, CVF-treated animals had less leukocytic infiltrate (CD45⁺) into the cornea at day 3 p.i. and, specifically, fewer infiltrating CSF1R⁺ cells (*Figure 5E,F*). By day 7 p.i., no difference in total CD45⁺ or CSF1R⁺ cells were observed in the cornea, yet there was a reduction in the total number of infiltrating CD4⁺ T cells (*Figure 5G*). Despite the difference in CSF1R⁺ cell infiltration into the corneas at day 3 p.i., HSV-1 titers were similar in the corneas and TG comparing vehicle- and CVF-treated animals (*Figure 5—figure supplement 1A*). By day 7 p.i. CVF-treated animals had less virus in the corneas and TG than vehicle controls (*Figure 5—figure supplement 1B*), suggesting that CVF may have unexpected antiviral effects. However, CVF treatment did not have any discernable impact on T cell expansion in the eye-draining MLN or the total number of circulating CXCR3⁺ CD4⁺ T cells in HSV-1 infected animals (*Figure 5—figure supplement 2*). Collectively, these findings imply that complement-targeted therapeutics could be tailored for ophthalmic use to limit tissue inflammation and preserve corneal sensation.

## Corneal sensation in other T cell-dependent ocular surface inflammatory diseases

While complement C3 and antigen-specific CD4⁺ T cells appear to coordinate corneal sensation loss in herpetic keratitis, it remained to be determined whether this phenomenon was specific to HSV-1 infection. To this end, corneal sensation was measured independent of infection in two T cell-dependent ocular surface inflammatory diseases. Murine models of allergic eye disease (AED) and ocular graft-versus-host disease (GVHD) were employed to further delineate the hypothesized role of T cells in coordinating corneal nerve damage (*Herretes et al., 2015*; *Lee et al., 2015*). First, AED was induced using an established systemic ovalbumin (OVA)-sensitization and ocular challenge model (*Figure 6A*) mimicking aspects of chronic keratoconjunctivitis (*Ahadome et al., 2016*; *Lee et al., 2015*). Clinical signs of ocular allergy developed as anticipated in OVA-challenged mice (*Figure 6B*), yet corneal sensation loss was not observed (*Figure 6C*). Consistent with development of AED, corneas from OVA-challenged mice exhibited CD3⁺ T cell infiltration by challenge day 7 (*Figure 6D*). However, frank corneal nerve loss was not observed in AED (*Figure 6D*). Morphometric analysis of confocal image stacks confirmed the preservation of corneal nerve density during AED (*Figure 6E*),

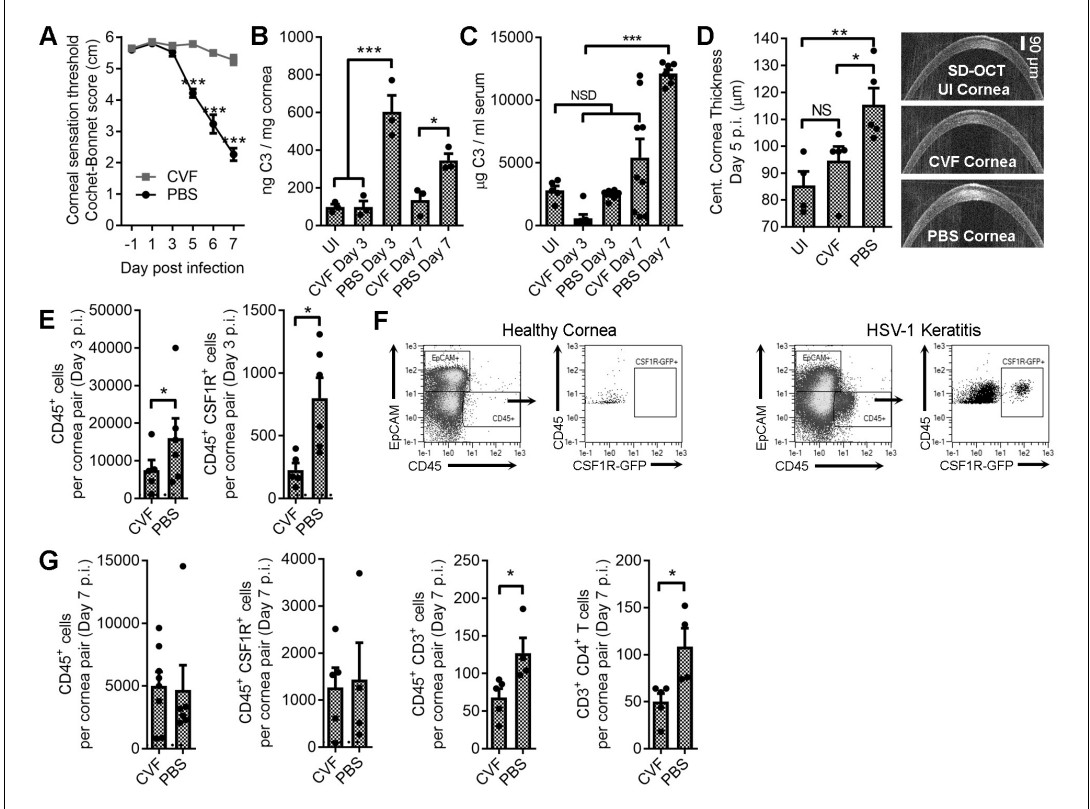

**Figure 5.** Local C3 depletion prevents HSV-associated corneal sensation loss. B6 mice were given PBS (vehicle) or 5.0 μg cobra venom factor (CVF) via subconjunctival injection to degrade C3, and ocularly infected with HSV-1 18 hr later. C3 depletion was maintained by daily topical treatment (eyedrop) containing 0.5 μg CVF. (A) Corneal sensation following HSV-1 infection in animals treated with CVF or PBS ($n$ = 5–11 mice/group; three independent experiments; two-way ANOVA, Bonferroni). Impact of CVF treatment on C3 protein concentrations in the cornea (B) and serum (C) ($n$= 3 cornea pairs, 5–9 serum samples/timepoint; 2–3 independent experiments; one-way ANOVA, Tukey). (D) Corneal edema measurements (central corneal thickness) determined via spectral domain optical coherence tomography (SD-OCT) on uninfected (UI) or HSV-1 infected mice treated with CVF or PBS at day 5 p.i. ($n$ = 4–5 mice/group; two experiments; one-way ANOVA, Newman-Keuls). (E) Impact of CVF treatment on total leukocyte (CD45+) and monocyte/macrophage (CSF1R+) infiltration into the corneas of CVF and PBS-treated MaFIA (CSF1R-GFP) mice at day 3 p.i. ($n$ = 5–6 mice/group; two independent experiments; Student's T). (F) Representative flow plots showing cell populations in healthy and HSV-1 corneas from panel E. (G) Impact of CVF treatment on leukocyte infiltration into the corneas of HSV-1 infected mice at day 7 p.i. ($n$ = 4–5 mice/group; two independent experiments; Student's T). Note: Total CD45+ graph in panel G reflects data from two technical replicates. Dashed lines in panels E and G reflect cell number recorded in healthy uninfected mice.

DOI: https://doi.org/10.7554/eLife.48378.013

The following figure supplements are available for figure 5:

**Figure supplement 1.** Impact of ocular cobra venom factor treatment on HSV-1 titers.

DOI: https://doi.org/10.7554/eLife.48378.014

**Figure supplement 2.** Immunologic impacts of ocular cobra venom factor treatment.

DOI: https://doi.org/10.7554/eLife.48378.015

although CD3+ T cell numbers were elevated in sensitized animals relative to healthy controls (*Figure 6F*). Flow cytometry was used to verify that the cornea-infiltrating T cells were predominately CD4+ in the AED model (*Figure 6G*). Although clinical data show that corneal nerve remodeling can occur in patients with chronic ocular allergy (*Hu et al., 2008*; *Le et al., 2011*), allergy symptoms were not accompanied by frank loss of corneal mechano-sensation in the AED animal model.

Corneal sensation was also evaluated in a T cell-dependent allogeneic model of chronic GVHD with systemic and ocular manifestations following hematopoietic stem cell (HSC) transplantation (*Perez et al., 2016*). This minor histocompatibility antigen mismatch model was established by transferring HSC-rich bone marrow (BM) and purified T cells from C57BL/6 (H2b) donors into lethally irradiated sex-matched C3.SW-H2b recipients. Recipient controls that receive BM without T cells

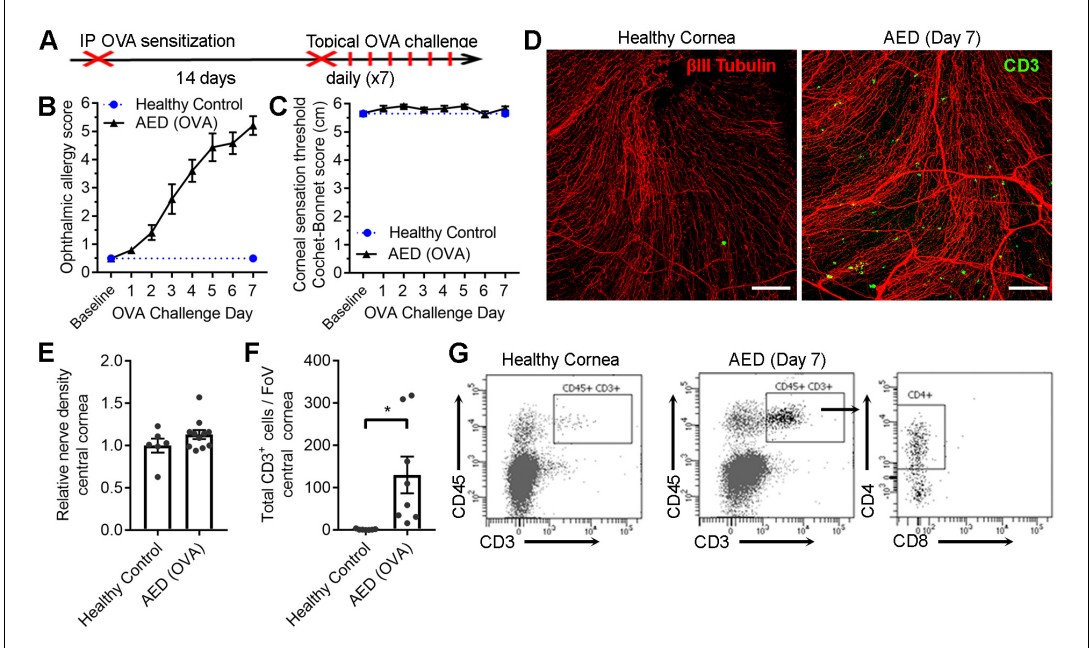

**Figure 6.** Chronic ocular allergy does not provoke frank corneal sensation loss. (**A**) Schematic of allergic eye disease induced by ovalbumin (OVA) immunization followed by topical ocular OVA challenge. Healthy and OVA-challenged B6 mice were evaluated for signs of ophthalmic allergy (**B**) and corneal sensation thresholds (**C**). (**D**) Representative confocal images of cornea flat mounts from healthy and OVA-challenged mice showing nerves (βIII Tubulin, red) and infiltrating T cells (CD3, green) in the central cornea at 20x magnification (scale bar = 100 µm). (**E**) Morphometric analysis of nerve densities in the central cornea based on confocal images. Normalization is based on the average volume in the healthy control group. (**F**) Quantification of total CD3+ cells per field of view. (**G**) Flow cytometry was used to confirm CD4 T cell infiltration into the corneas of OVA-challenged mice. Data in panels B – F reflect 5 to 6 mice per group; flow plots in panel G reflect pooled digests of corneas from three mice. Data in panels E and F were analyzed using Student's T tests.

DOI: https://doi.org/10.7554/eLife.48378.016

recover and do not develop GVHD (*Figure 7A*). Onset of GVHD was established by transfer of BM with CD4+ T cells only, CD8+ T cells only, or both CD4+ and CD8+ T cells. Progressive corneal sensation loss was observed following transfer of CD4+ T cells with and without addition of CD8+ T cells. Addition of CD8+ T cells alone evoked a transient corneal sensation deficit that recovered by day 26 post-transplant. However, corneal sensation loss was not observed in the BM only control group (*Figure 7B*). External signs of ocular disease (see *Table 2*) were also apparent in groups receiving CD4+ T cells by the study endpoint (*Figure 7C*). The presence of tissue-infiltrating CD3+ T cells in corneas from animals with GVHD was verified by confocal microscopy at the study endpoint. Consistent with sensation loss, corneal nerve integrity was markedly reduced in groups receiving CD4+ T cells (*Figure 7D*). Morphometric analysis of confocal image stacks confirmed the CD4-dependent decrease in corneal nerve density (*Figure 7E*) and concomitant increase in total CD3+ cells (*Figure 7F*). Taken together, our data show for the first time that corneal sensation loss occurs early in ocular GVHD and that this pathology is instigated in part by allogeneic CD4+ T cells.

## Localized pharmacologic C3 depletion preserves corneal sensation in ocular GVHD

The pharmacologic impact of local CVF treatment on ocular GVHD progression was consequently explored in order to identify whether the CD4+ T cell-associated corneal sensation loss observed in both herpetic keratitis and ocular GVHD shared a common C3-codependent pathomechanism. For these experiments, GVHD was established in C3.SW-H2b recipients by transfer of C57BL/6-derived BM and CD3+ T cells. Recipients from the BM only control and GVHD cohorts received ophthalmic treatment with either PBS or CVF (*Figure 8A*). Systemic disease progressed equally in both GVHD cohorts regardless of treatment (*Figure 8B*). Although CVF administration was limited to the eye,

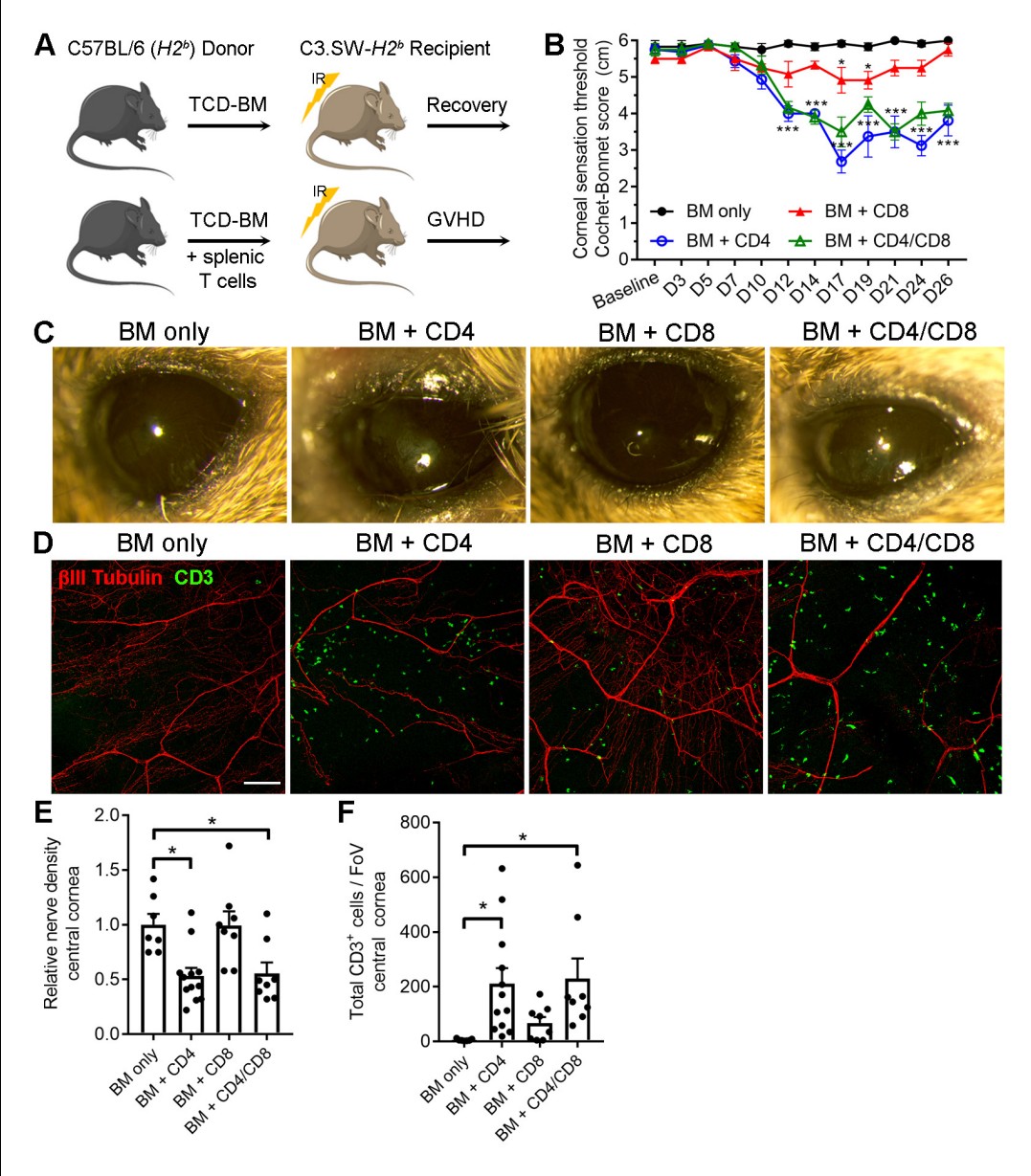

**Figure 7.** CD4 T cells drive corneal sensation loss in ocular GVHD. (A) Schematic of GVHD induction using T cell depleted bone marrow (TCD-BM) or TCD-BM with splenic T cells isolated from C57BL/6 donors and transferred into C3.SW-H2 b recipients. (B) Corneal sensation measurements in C3.SW-H2 b mice following reconstitution with BM only or with BM and 1.3x10 6 CD4, CD8, or CD4 and CD8 T cells. (C) External photographs showing CD4-dependent ocular surface morbidities consistent with corneal sensation loss. (D) Representative confocal images of cornea flat mounts from each group of mice at the study endpoint showing nerves (βIII Tubulin, red) and infiltrating T cells (CD3, green) in the central cornea at 20x magnification (scale bar = 100 µm). (E) Morphometric analysis of nerve densities based on confocal images. Normalization is based on the average volume in the BM only control group. (F) Quantification of total CD3 + cells per field of view. Data reflect independent measurements of corneas from 3-4 mice per group across 2 experiments; sensation data were evaluated by two-way ANOVA with Bonferroni posttests; data in panels E and F were analyzed by one-way ANOVA with Bonferroni posttests. Figure was generated using Servier Medical Art (http://smart.servier.com/) under a Creative Commons 3.0 license.

DOI: https://doi.org/10.7554/eLife.48378.017

systemic serum C3 concentrations were lower in the GVHD cohort treated with CVF compared to the PBS-treated GVHD groups at the study endpoint. Nonetheless, the inverted CD4:CD8 T cell ratio (*Herretes et al., 2015*) in the spleens and eye-draining mandibular lymph nodes of both groups of GVHD mice confirmed the presence of systemic disease (*Figure 8D*). Despite development of systemic disease, local CVF treatment preserved corneal nerve integrity and mechano-sensory function in ocular GVHD (*Figure 8E,F*). Morphometric analysis of confocal image stacks from the study endpoint corroborated the preservation of corneal nerve density in CVF-treated mice from the GVHD cohort (*Figure 8G*), although CVF did not reduce the number of total CD3[+] cells during GVHD relative to PBS (*Figure 8H*). In addition, CVF treatment also limited GVHD-associated periocular disease in female but not in male mice (*Figure 8—figure supplement 1*). In summary, these preclinical models establish that CD4[+] T cells and complement C3 coordinate corneal sensory nerve damage in both herpetic keratitis and ocular GHVD. Moreover, our data provide a proof of principle that complement-targeted therapeutics may limit the severity of immune-mediated sensory nerve damage at the ocular surface.

## Discussion

Mechanistic advances in our understanding of the complement cascade's physiologic regulation and pathologic contributions in disease have predictably spawned substantial investments in complement-targeted drug development over the past decade (*Harris et al., 2018*; *Ricklin et al., 2018*; *Tomlinson and Thurman, 2018*). The ocular surface is a unique milieu in which pharmacologic modulation of the complement pathway may limit the severity of inflammatory disease and improve clinical outcomes. This study provides proof of concept that such interventions may have important clinical impacts on ocular surface disease, and specifically neuropathic sensory disorders affecting the cornea. Complement-targeted drug development for ophthalmic use has almost exclusively focused on AMD with some interest in neuromyelitis optica (NMO), Stargardt disease, and autoimmune uveitis (*Harris et al., 2018*; *Pittock et al., 2013*). By July 2019, there were no complement-specific therapeutics indicated for ocular surface use registered on www.clinicaltrials.gov (search strings: 'complement' / 'C3' / 'C5' / 'eculizumab' AND 'cornea', 'conjunctiva', 'dry eye', 'keratitis', OR 'ocular surface'). Nonetheless, the complement pathway is implicated in the pathophysiology of several ocular surface diseases (*Bora et al., 2008*). The current investigation provides an important advancement in this arena by demonstrating that dysregulated complement activation can specifically contribute to sensory nerve damage in the cornea.

While corneal nerves are increasingly implicated in maintenance of corneal immune privilege (*Paunicka et al., 2015*; *Neelam et al., 2018*; *Guzmán et al., 2018*), this study is the first to identify that C3 is involved in the pathobiology of corneal sensory nerve damage. However, the individual fates of damaged corneal nerves were not explored herein. Lessons from peripheral nerve injury in other settings indicate that some damaged nerves undergo apoptosis, yet effectively promoting

**Table 2.** Ophthalmic Disease Scoring Guide for Ocular GVHD.

| Criteria | Grade 0 | Grade 1 | Grade 2 | Grade 3 |
|---|---|---|---|---|
| Eyelid Closure | None | Mild squinting | Partial lid closure | Full lid closure |
| Blepharitis | None | Mild lid margin edema | Moderate lid margin edema with crusting | Gross lid edema, loss of lashes/ eyelid fur |
| Meibomian Gland Disease | None | ≤4 plugs | >4 plugs OR cystic plugs | >4 plugs AND cystic plugs |
| Conjunctival Chemosis | None | Mild edema | Moderate edema | Gross edema |
| Conjunctival Hyperemia | None | Mild redness | Broad erythema | N/A |
| Mucoid Discharge | None | Mild discharge | Discharge covers the ocular surface | N/A |
| Corneal Opacity | None | Epithelial haze | Focal opacities | Widespread opacity |

DOI: https://doi.org/10.7554/eLife.48378.018

functional regeneration of surviving neurons remains a major clinical hurtle (*Doron-Mandel et al., 2015*; *Menorca et al., 2013*; *Scheib and Höke, 2013*; *Shacham-Silverberg et al., 2018*). Likewise, promoting functional regeneration of damaged corneal nerves is important for restoration of immune privilege and ocular surface health. Our approach evaluated sensory nerve damage in murine models of T cell-dependent ocular surface disease including HSV-1 keratitis, OVA-induced AED, and ocular GHVD. The lack of a sensory phenotype in the AED model was unanticipated in light of evidence that patients with severe allergy exhibit changes in corneal nerve morphology (*Hu et al., 2008*; *Le et al., 2011*; *Leonardi et al., 2012*). This contrasting model underscores the context-dependency of sensation loss during corneal inflammation.

While we focused on C3 in HSV-1 keratitis and ocular GVHD due to the direct link to overt corneal sensation loss, complement may still have an important role in mediating neurogenic hypersensitivities in AED. In addition to loss of mechano-sensory function, corneal nerve pathologies can involve a broad array of neuropathic clinical symptoms including dryness, itch, and pain (*Andersen et al., 2017*). Inflammatory mediators facilitate neuropathic hypersensitivities, but the therapeutic potential of complement inhibition is often overlooked (*Baral et al., 2019*; *Fritzinger and Benjamin, 2016*). Moreover, the cornea produces many neurotrophic factors that influence sensory innervation in health and disease (*Sacchetti and Lambiase, 2017*; *Yu et al., 2015*). Given the involvement of C3 activation in sensory nerve damage, future investigation into crosstalk between complement and other neurotropic factors in the cornea is warranted. Furthermore, AED is driven primarily by a combined Th2/Th17 CD4 T cell response that differs from the Th1 bias observed in HSV-1 keratitis and ocular GVHD (*Herretes et al., 2015*; *Saban et al., 2013*; *Tang and Hendricks, 1996*). This suggests that Th1 cytokines such as IFNγ, IL-2, and lymphotoxin-alpha may be important in coordinating corneal sensation loss. Nonetheless, the unique inflammatory milieus and differential polarizations of cornea-infiltrating T cells across these disease models may also influence the downstream effects of complement activation in the tissue microenvironment (*Tomlinson and Thurman, 2018*). Future studies are needed to broaden the scope of these observations to other peripheral nerve diseases. For example, complement-mediated microvascular damage in diabetes correlates with neuropathy (*Rasmussen et al., 2018*; *Rosoklija et al., 2000*). While diabetic peripheral neuropathy can lead to neurotropic corneal ulcerations (*Gao et al., 2016b*), the role of complement in diabetic corneal disease is currently unknown.

Complement enhances clearance of infected cells and nascent virions during HSV-1 infection (*Kostavasili et al., 1997*; *Lubinski et al., 2002*; *McNearney et al., 1987*). Importantly, HSV-1 can also evade C3 activation through endogenous expression of glycoprotein C (gC). By limiting C3b deposition on infected cells and nascent virions, gC can inhibit subsequent C5 binding and membrane attack complex (MAC) formation (*Kostavasili et al., 1997*; *Lubinski et al., 2002*; *McNearney et al., 1987*; *Tegla et al., 2011*). This could explain why herpetic keratitis is less severe in rabbits infected with a gC-deficient strain of HSV-1 during acute infection (*Drolet et al., 2004*). However, this gC-mediated viral evasion strategy may also be cell type-dependent. Cell culture modeling indicates that neuronal cells maintain host-encoded complement regulator expression more efficiently and are more resistant to MAC deposition than epithelial cells upon HSV-1 infection (*Rautemaa et al., 2002*). Accordingly, complement-mediated sensory nerve damage in HSV-1 keratitis may reflect a maladaptive outcome of host-defense. This is consistent with our data showing that immunologically naive C3$^{-/-}$ mice maintain corneal sensation and innervation, yet they exhibit concomitant enhancement of viral shedding in the tear film and increased viral burden in the TG by day 7 p.i. The absence of a concordant difference in HSV-1 titers in corneas from WT and C3$^{-/-}$ mice is likely explained by the potent antiviral effects of type-1 interferon (*Conrady et al., 2011*; *Royer and Carr, 2016*). This suggests that the maintenance of corneal innervation in C3$^{-/-}$ mice enables enhanced shedding of nascent virions produced within the TG. Moreover, there is no difference in the amount of latent HSV-1 in the TG of naive WT and C3$^{-/-}$ mice 30 days after ocular HSV-1 challenge (*Royer et al., 2019*). Taken together, this evidence indicates that although C3 is involved in progression of keratitis and sensory nerve fiber retraction during acute infection, it likely does not contribute to complete elimination of infected nerves.

The studies reported herein involve immunologically naive mice, but the impacts of preexisting humoral immune responses on complement pathway activation and regulation in the cornea during HSV-1 infection remain incompletely understood. This point is of considerable importance clinically, as herpes-associated neurotropic keratitis typically develops as a result of recurrent corneal

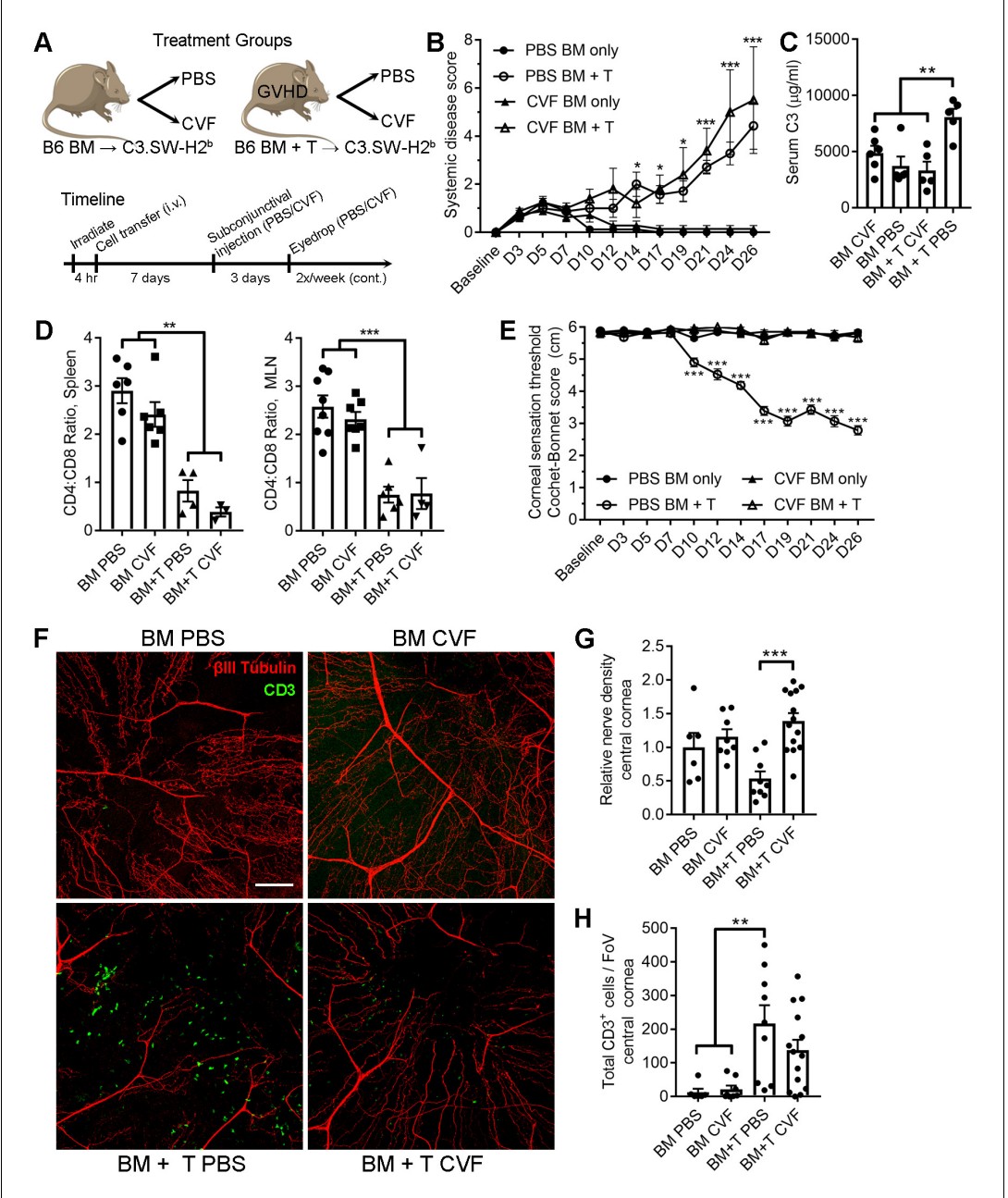

**Figure 8.** Local C3 depletion prevents corneal sensation loss in ocular GVHD. (**A**) Experiment schematic of GVHD induction using T cell depleted bone marrow (TCD-BM) or TCD-BM with splenic T cells isolated from C57BL/6 donors and transferred into C3.SW-H2 b recipients. Each group was subsequently given a subconjunctival injection containing PBS (vehicle) or 5.0 μg cobra venom factor (CVF) to degrade C3. C3 depletion was maintained by topical treatment (eyedrop) containing 2.0 μg CVF twice weekly. (**B**) Systemic disease scores in each cohort (n=8 mice per group; two-way ANOVA, Bonferroni). (**C**) Impact of CVF treatment on serum C3 protein concentrations in each group at the experiment endpoint (n=5-6 samples/group; one-way ANOVA, Bonferroni). (**D**) Evaluation of CD4:CD8 ratios in secondary lymphoid organs by flow cytometry to confirm onset of GVHD at experimental endpoints (n=3-8 mice/group; 2 independent experiments; one-way ANOVA, Bonferroni). (**E**) Longitudinal corneal sensation measurements in each group of C3.SW-H2 b mice. (**F**) Representative confocal images of cornea flat mounts from each group of C3.SW-H2 b mice at the study endpoint showing nerves (βIII Tubulin, red) and infiltrating T cells (CD3, green) in the central cornea at 20x magnification (scale bar = 100 μm). (**G**) Morphometric analysis of nerve densities based on confocal images. Normalization is based on the average volume in the BM only control group. (**H**) Quantification of total CD3 + cells per field of view. Data in panels E – H reflect 3-8 mice/group; 2 independent experiments; two-way ANOVA with Bonferroni posttests. Figure was generated using Servier Medical Art (http://smart.servier.com/) under a Creative Commons 3.0 license.
DOI: https://doi.org/10.7554/eLife.48378.019

The following figure supplement is available for figure 8:

*Figure 8 continued on next page*

*Figure 8 continued*

**Figure supplement 1.** Sex-biased effects of local C3 depletion on ocular GVHD severity.

DOI: https://doi.org/10.7554/eLife.48378.020

infections in patients (*Hamrah et al., 2010*). Furthermore, we have recently reported that C3 is essential for optimal antibody-dependent viral clearance following ocular HSV-1 challenge in mice (*Royer et al., 2019*; *Royer et al., 2017*). While vaccinated animals did not exhibit corneal sensation loss following ocular HSV-1 infection in those studies, deposition of the terminal C3 cleavage product C3d was present in the corneal epithelium (*Royer et al., 2017*). Host gene expression data herein corroborates that HSV-1 infection creates an imbalance in the local complement effector to regulator expression ratios that contribute to aberrant complement pathway activation in the cornea. This imbalance in 'complement proteostasis' may favor deposition of complement fragments and sublytic MAC on corneal sensory nerve fibers (*Tegla et al., 2011*; *Triantafilou et al., 2013*). Notwithstanding, the nerve-intrinsic molecular mechanisms responsible for trigeminal sensory fiber retraction, neuronal death, or axonal regeneration in the cornea are largely unknown (*Stepp et al., 2017*). Future work is needed to identify the respective complement activation pathways involved in pathologic and protective immune responses to HSV-1 in the cornea.

Identification of the complement pathway's role in initiating corneal sensation loss following HSV-1 infection is an important advancement in the mechanistic understanding of this disease process, as it introduces an array of potential drug targets for local therapy. Previous observations note that complement activation in antigen-induced keratitis may be mediated by cellular immune mechanisms (*Verhagen et al., 1992*). The identification of CSF1R$^+$ macrophages/monocytes as a local source of C3 during infection (*Figure 4D*) is also critical, as these cells are recruited to the cornea in the early stages of HSV-1 infection and are associated with corneal nerve damage in HSV-1 keratitis (*Chucair-Elliott et al., 2017b*; *Conrady et al., 2013*). Moreover, CD4 T cells coordinate this pathology. Our data corroborate findings from the Hendricks' lab showing that CD4 T cells are associated with corneal sensation loss in HSV-1 keratitis (*Yun et al., 2014*), yet our adoptive transfer experiments establish that this process is not dependent upon endogenous C3 production by cornea-infiltrating donor T cells (*Figure 2D*). Notably, the tempo of HSV-associated sensation loss was consistent among models once initiated (*Figure 1B*, *Figure 2D*, *Figure 3A*). However, the initial onset was delayed by one day in the adoptive transfer model. This likely reflects the relative lymphopenic status of T cell-reconstituted TCRα$^{-/-}$ mice compared to WT mice (compare *Figure 2B* and *Figure 3—figure supplement 1*).

Corneal sensation loss in HSV-1 infection and ocular GVHD shared a common mechanism dependent upon CD4$^+$ T cells and complement C3. The role of intracellular C3 in regulating human Th1 responses has received much attention in recent years (*Elvington et al., 2017*; *Hansen et al., 2019*; *Liszewski et al., 2013*). Such studies involve CD3 and CD46 stimulation to activate human CD4 T cells in vitro. However, mice do not express CD46. Instead, anaphylatoxin receptor signaling has been shown to modulate Th1 cytokine production in murine T cells (*Strainic et al., 2008*). Our ex vivo re-stimulation data clearly show that there is no defect in IFNγ production by CD4 T cells from C3$^{-/-}$ mice following HSV-1 infection. Others have shown similar data based on T cell proliferation in response to re-stimulation with HSV-1 antigen (*Da Costa et al., 1999*). In contrast, a reduction in IFNγ production by CD4 T cells from C3$^{-/-}$ mice was reported following in vitro expansion under Th1 polarizing conditions (*Liszewski et al., 2013*). Nonetheless, it is possible that anaphylatoxin receptor signaling from locally produced C3a (and potentially C5a generated downstream) modulates T cell effector function upon extravasation into the inflamed cornea. The precise relationships between T cells and C3 in our mouse models of corneal neuropathic disease remain to be identified. Our working hypothesis involves an indirect mechanism whereby sublytic MAC deposition from local complement activation damages sensory nerves, which are then cleared by T cell-activated phagocytes or other cytolytic cells. These could include tissue-resident and infiltrating leukocytes (macrophages, monocytes, dendritic cells, and NK cells) as well as nerve-associated corneal epithelial cells (*Buela and Hendricks, 2015*; *Chucair-Elliott et al., 2017b*; *Gao et al., 2016a*; *Koyama and Hill, 2016*; *Liu et al., 2017*; *Royer et al., 2018*; *Seyed-Razavi et al., 2014*; *Stepp et al., 2017*). Emerging evidence indicates that the plasma membranes of corneal nerves and epithelial cells fuse thereby

enabling cytosolic exchange (*Stepp et al., 2017*). In light of this, it is plausible that complement-mediated damage to corneal epithelial cells has a direct impact on sensory nerves.

Although corneal nerve damage is an established pathology in animal models of HSV-1 infection (*Chucair-Elliott et al., 2015*; *He et al., 2017b*; *Yun et al., 2014*), this is the first report documenting loss of corneal mechano-sensory function in ocular GVHD. Notably, patients rarely exhibit neurological manifestations of GVHD affecting other tissues/organs (*Grauer et al., 2010*). Corneal sensation loss may provide a clinical benchmark for initiation of targeted therapies to curb progression of ocular GVHD in patients. Data from our animal model of GVHD suggest that corneal sensation loss is an early warning sign of progressive ocular and systemic GVHD. Corneal sensation measurements are warranted in patients following HSCT to substantiate this finding among those who develop chronic GVHD. Published imaging studies confirm that corneal nerve remodeling occurs in patients with ocular GVHD (*He et al., 2017a*; *Tepelus et al., 2017*); therefore, clinical studies to delineate the kinetics of disease onset are needed. Complement C3 is a known driver of systemic GVHD (*Kwan et al., 2012*; *Ma et al., 2014*; *Seignez et al., 2017*), and our data show that aberrant complement pathway activation also contributes to the pathogenesis of ocular GVHD including corneal sensation loss. Furthermore, localized CVF treatment preserved corneal sensation in both sexes during ocular GVHD, but treatment only abrogated other facets of ocular surface disease in female mice. This dimorphic outcome may reflect insufficient maintenance of C3 depletion in the ocular surface microenvironment, as animals were only treated twice weekly. Furthermore, complement activity is reportedly limited by terminal pathway components in female mice compared to males (*Kotimaa et al., 2016*). Collectively, our data suggest that sensory nerve involvement may be a unique facet and treatment target in ocular GVHD.

While the complement pathway has many components, C3 is the heart of the pathway and its activation products mediate virtually all downstream pathway functions. Accordingly, we used CVF to target complement activation in models of HSV-1 keratitis and ocular GVHD as a proof of concept for local delivery of complement-targeted therapeutics for ocular surface disease. In contrast to the neurotoxic effects of complete cobra venom, purified CVF is a 'nontoxic' derivative. CVF forms a biochemically stable convertase to rapidly hydrolyze mammalian C3 and C5. This ultimately results in complement inactivation via effector consumption (*Vogel and Fritzinger, 2010*). The only recorded side effect of purified CVF administration in rodents is transitory anaphylatoxin-mediated pulmonary inflammation resulting in acute respiratory distress (*Proctor et al., 2006*; *Vogel and Fritzinger, 2010*). In our hands, respiratory distress was observed in some mice following the initial sub-conjunctival CVF injection irrespective of group assignment (animals were euthanized). However, topical CVF administration did not provoke respiratory or corneal abnormalities in any animals. Previous findings also show that topical CVF administration has no obvious effects on corneal health or transparency (*Zaidi et al., 2010*).

Experimental CVF administration was once thought to directly stimulate T cells, but such effects were later attributed to mitogen contamination (*Rumjanek et al., 1978*; *Cauvi et al., 2012*). Because our disease models were T cell-dependent, we utilized highly-purified CVF from a commercial source for our studies. Moreover, we found no evidence to suggest that CVF impacted T cell responses outside of the eye. Local CVF treatment reduced corneal T cell infiltration following HSV-1 infection (*Figure 5G*). However, this is likely due to reduced inflammation overall in C3-depleted corneas (*Figure 5D*). Likewise, ocular CVF treatment in the GVHD model did not impact on the total numbers of T cells in the corneas at the experiment endpoint. However, we have not excluded the possibility that CVF treatment may delay the tempo of T cell influx. Nonetheless, CVF administration preserved corneal sensation in both disease models. By addressing the 'heart' of the complement pathway (C3), this study opens the door to future investigation to further elucidate the relevant pathomechanisms and corresponding therapeutic targets. The relevant complement activation pathways (classical, alternative, lectin, terminal), cellular targets, and impacts of anaphylatoxin receptor signaling are topics of future investigation.

Complement activation is a double-edged sword in the cornea, as activation can be either protective or harmful. Low-level complement turnover is observed in healthy corneas/tears, and complement may even contribute to the relative 'paucibacterial' state of the ocular surface microbiome (*Doan et al., 2016*; *McDermott, 2013*; *Willcox et al., 1997*). The mechanisms regulating each complement-mediated effect remain incompletely understood. Elucidation of how complement activation is triggered, the relevant molecular mechanisms of its downstream signaling, and contributions

to pathology will promote an important advancement in drug development for ocular surface disease. Animal models are necessary for mechanistic studies when it comes to understanding complement pathway regulation, especially in dynamic tissue microenvironments; however, differences in complement regulation exist between mouse and man that will also have to be resolved to further assess the translational potential (*Jacobson and Weis, 2008*). This certainly applies to putative complement-targeted therapeutics for ocular surface treatment. Even among individual patients, small nucleotide polymorphisms evoke changes in complement activity that have major impacts on disease risks. Such genetic linkages, dubbed 'complotype,' are important in differential susceptibility to AMD (*Harris et al., 2012*; *Paun et al., 2016*). Accordingly, complotype differences may help explain differential susceptibility patterns in a variety of ocular surface diseases. Our data herein have demonstrated successful prophylactic intervention in repressing complement-mediated sensory nerve pathology in two well characterized experimental T cell-dependent corneal disease models. Other avenues of future investigation are also needed to determine the efficacy of therapeutic intervention on established ocular surface diseases. Successful implementation of complement-targeted therapeutics for topical ophthalmic use may provide the benefit of controlling insidious ocular surface diseases without the risks associated with systemic therapies.

# Materials and methods

**Key resources table**

| Reagent type (species) or resource | Designation | Source or reference | Identifiers | Additional information |
|---|---|---|---|---|
| Strain, strain background (*Mus musculus*, C57BL/6J) | Wildtype C57BL/6 (WT) | Jackson Laboratories | Stock # 000664 | |
| Strain, strain background (*M. musculus*, C57BL/6J) | Complement C3 deficient (C3-/-) | Jackson Laboratories | Stock # 029661 | |
| Strain, strain background (*M. musculus*, C57BL/6J) | T cell receptor alpha deficient (TCRα-/-) | Jackson Laboratories | Stock # 002116 | |
| Strain, strain background (*M. musculus*, C57BL/6J) | Transgenic OVA-specific CD4 T cells (OT-II) | Jackson Laboratories | Stock # 004194 | |
| Strain, strain background (*M. musculus*, C57BL/6J) | Transgenic Csf1r-eGFP (MAFIA) | Jackson Laboratories | Stock # 005070 | |
| Strain, strain background (*M. musculus*, C57BL/6J) | C3.SW-H2b | Jackson Laboratories | Stock # 000438 | |
| Strain, strain background (Herpes simplex virus type 1, McKrae) | HSV-1 | *Macdonald et al., 2012*; *Watson et al., 2012* | N/A | Originally from Brian Gebhardt |
| Cell line (*Cercopithecus aethiops*) | Vero | American Type Culture Collection (ATCC) | Cat. # CCL-81 | see Materials and methods |
| Chemical compound, drug | Cobra Venom Factor, *Naja naja kaouthia* | Millipore-Sigma | Cat. # 233552 | |
| Commercial assay or kit | Anti-mouse CD90.2 IMag particles | BD | Cat. # 551518 | |

*Continued on next page*

*Continued*

| Reagent type (species) or resource | Designation | Source or reference | Identifiers | Additional information |
|---|---|---|---|---|
| Commercial assay or kit | Anti-mouse CD4 microbeads | Miltenyi Biotec | Cat. # 130-049-201 | |
| Commercial assay or kit | Anti-mouse CD8 microbeads | Miltenyi Biotec | Cat. # 130-116-478 | |
| Commercial assay or kit | Anti-mouse CD90.2 microbeads | Miltenyi Biotec | Cat. # 130-049-101 | |
| Commercial assay or kit | Anti-mouse CD45 microbeads | Miltenyi Biotec | Cat. # 130-052-301 | |
| Commercial assay or kit | Anti-mouse EpCAM microbeads | Miltenyi Biotec | Cat. # 130-105-958 | |
| Antibody | Rabbit polyclonal IgG (anti-mouse beta-III tubulin) | Abcam | Cat. # ab18207 | (1:500) |
| Antibody | AlexaFluor647 donkey polyclonal IgG (anti-rabbit IgG) | Jackson Immunoresearch | Cat. # 711-605-152 | (1:1000) |
| Antibody | Anti-mouse CD3e FITC (Clone: 145–2 C11) | eBioscience (ThermoFisher) | Cat. # 11-0031-82 | (1:500) |
| Other | Roche Liberase TL | Sigma Aldrich | Cat. # 5401020001 | see Materials and methods |
| Commercial assay or kit | C3 ELISA | Abcam | Cat. # ab157711 | |
| Commercial assay or kit | Milliplex MAP Luminex Array | EMD Millipore | Custom Design | |
| Recombinant DNA reagent | iScript cDNA synthesis kit | Biorad | Cat. # 1708891 | |
| Recombinant DNA reagent | Sso Advanced SYBR Green qPCR Supermix | Biorad | Cat. # 1725270 | |
| Other | Trizol | ThermoFisher | Cat. # 15596026 | see Materials and methods |
| Software, algorithm | Prism 6 | GraphPad | N/A | |
| Software, algorithm | Imaris x64 (8.2.1) | Bitplane | N/A | |

## Animal models

Mouse strains were originally purchased from The Jackson Laboratory (Bar Harbor, ME). These include: C57BL/6, C3$^{-/-}$ (stock #029661), TCR$\alpha^{-/-}$ (stock # 002116), OT-II (stock #004194), MAFIA/CSF1R-GFP (stock #005070), C3.SW-H2$^{b}$ (stock #000438), and B6-GFP (stock #003291). Colonies were maintained in select-pathogen-free vivaria at the University of Oklahoma Health Sciences Center or Duke University Medical Center. Research was performed in accordance with protocols approved by respective institutional animal care and use committees.

For all procedures relating to ocular HSV-1 infection, C57BL/6-background mice were anesthetized with ketamine and xylazine and euthanized by cardiac perfusion (*Royer et al., 2018*). For infection, corneas were scratched to expose the epithelium and 1000 plaque forming units (PFU) of HSV-1 McKrae was applied to each eye. Viral stocks were propagated and infectious titers quantified by standard plaque assay on Vero cells (American Type Culture Collection, Manassas, VA) as previously reported (*Royer et al., 2015*). For adoptive transfers, T cells were purified from splenocyte preparations using a BioRad S3e cell sorter (Hercules, CA). Corneal complement C3 depletion was achieved by subconjunctival injection of CVF (Cat. # 233552, Millipore Sigma, Burlington, MA) followed by topical maintenance dosing as described in respective figure legends.

Allergic eye disease was initiated in B6 mice as previously described (*Ahadome et al., 2016*; *Lee et al., 2015*; *Schlereth et al., 2012*). Briefly, hypersensitivity was elicited via intraperitoneal

immunization with 10 µg ovalbumin (OVA) adjuvanted with 1 mg Alum and 300 ng pertussis toxin (all from Sigma-Aldrich Corp., St. Louis, MO). Ocular allergy was subsequently induced via daily eye-drop challenge containing 250 µg OVA (*Figure 6A*). Clinical signs of AED were scored daily using established criteria (*Schlereth et al., 2012*). Animals were euthanized by $CO_2$ asphyxiation for tissue collection.

Induction of GVHD was based on an established T-dependent, MHC-matched unrelated donor model (*Perez et al., 2016*). Briefly, C3-SW.H2$^b$ mice were lethally irradiated (10.5 Gy, Cs-137) and reconstituted with $5 \times 10^6$ T cell-depleted bone marrow (TCD-BM) cells from allogeneic C57BL/6 (H2$^b$) mice. T cell depletion was achieved using BD anti-mouse CD90.2 IMag particles (Cat. # 551518, San Jose, CA). Control groups received TCD-BM only, but GVHD onset required reconstitution with TCD-BM and T cells. T cells were isolated from B6 splenocytes using immunomagnetic microbeads (Miltenyi Biotec, San Diego, CA) targeting CD4 (Cat. #130-049-201), CD8 (Cat. #130-116-478), or CD90.2 (Cat. #130-049-101) according to the manufacturer's instructions. Cell purity was evaluated by flow cytometry (~80%) and adjusted such that $1.3 \times 10^6$ CD4$^+$ or CD8$^+$ T cells or $2.3 \times 10^6$ CD3$^+$ T cells were injected into each recipient with TCD-BM. Irradiated animals received water containing gentamicin (6.6 µg/ml) for ten days following irradiation (Henry Schein Animal Health, Dublin,OH). Supplical nutritional paste (Henry Schein) was also provided ad libitum to reduce weight loss. Systemic manifestations of GVHD were scored using established criteria (*Perez et al., 2016*) and ocular disease scored according to *Table 2*.

## Corneal nerve mechano-sensation, imaging, and morphometric analysis

Corneal mechano-sensory function was measured on non-anesthetized mice with a Luneau Cochet-Bonnet esthesiometer (Western Ophthalmics, Lynwood, WA) in 0.5 cm increments on a scale of 0 to 6 cm as previously described (*Chucair-Elliott et al., 2015*). Sensation scores reflect the longest filament length capable of eliciting replicate blink reflexes when applied to the central cornea. A Bioptigen spectral domain optical coherence tomography (SD-OCT) system (Leica Microsystems, Buffalo Grove, IL) was utilized to evaluate corneal inflammation and edema in vivo (*Downie et al., 2014*). Digital photography of the external eye was captured with a Leica MZ16 FA stereomicroscope.

Corneal nerves were imaged in flat-mounted corneal buttons as described (*Chucair-Elliott et al., 2015*). Briefly, corneas were harvested with the limbus intact, fixed, permeabilized, and labeled with Abcam anti-mouse beta-III tubulin primary antibody (Cat. # ab18207, Cambridge, MA) and a corresponding AlexaFluor647-conjugated secondary antibody (Cat. # 711-605-152, Jackson Immunoresearch, West Grove, PA). Corneas were also labeled with an eBioscience FITC-conjugated anti-mouse CD3ε antibody (ThermoFisher, Waltham, MA). Slides were imaged using an Olympus Fluoview1200 laser scanning confocal (Center Valley, PA) or a Nikon AR1 HD resonant scanning confocal (Melville, NY) microscope with sequential channel scanning.

Confocal z-stack images were evaluated using Imaris software (Bitplane, Concord, MA) to quantify corneal nerves densities and the total numbers of CD3$^+$ cells. Briefly, images were displayed as maximum intensity projections. Nerve volume per field of view was measured with the 'create surface tool' with manual thresholding for voxel area coverage of βIII tubulin$^+$ fibers (default parameters; smooth surface area detail set to 0.5 µm; background subtraction set to 8.0 µm). Total CD3$^+$ cells were quantified using the spots tool with default parameters, an estimated cell diameter of 8 µm, and classification using the quality filter method with manual thresholding. Morphometric nerve volume and cell count summary statistics were reported by the software per image (20x field of view). Nerve densities were normalized to the respective control group average.

## Flow cytometry and cell isolation

Secondary lymphoid organs were harvested and mechanically processed into single-cell suspensions (*Royer et al., 2015*). Blood was collected from the facial vein and treated with erythrocyte lysis buffer prior to labeling. Corneas were digested in RPMI1640 media containing 0.26 Wünsch units of Roche LiberaseTL enzyme blend (Sigma Aldrich) at 37° C as described (*Royer et al., 2017*). Cell suspensions were filtered and labeled with Fc block and fluorochrome-conjugated antibodies (eBioscience). In vitro T cell functional assays were performed as previously described (*Royer et al., 2015*). Briefly, cells were stimulated using 50 ng PMA and 800 ng ionomycin for three hours and monensin added after one hour (BD Biosciences). Intracellular IFNγ expression was evaluated by

flow cytometry using a BD cytofix/cytoperm staining kit. Samples were analyzed on a Miltenyi Biotec MacsQuant10 or a BD LSRFortessa flow cytometer with MacsQuantify or FacsDiva software, respectively. For downstream gene expression studies, cornea digests were fractionated with Miltenyi microbeads by sequentially targeting CD45 (Cat. # 130-052-301) and EpCAM (Cat. # 130-105-958) as described (Royer et al., 2018). Alternatively, GFP-expressing cells were purified or depleted using a BioRad S3e cell sorter.

### Immunoassays and gene expression

Tissues were harvested and homogenized in phosphate-buffered saline (PBS) containing 1x Calbiochem protease inhibitor cocktail (Millipore Sigma). Serum was obtained by collecting blood from the facial vein into BD microtainer serum separator tubes. Tissue homogenates and serum were clarified by centrifugation at $10,000 \times g$. Chemokine concentrations were determined using a BioRad Luminex system and Millipore Milliplex MAP technology. Concentrations of C3 were measured by ELISA (Abcam, Cat. # ab157711).

Gene expression studies were performed using RNA isolated from tissue and cells using the Trizol (ThermoFisher) method and converted to cDNA with iScript (BioRad). Real-time PCR was performed using PrimePCR technology with commercially validated primer sequences (Biorad) and a Biorad CFX-Connect thermocycler as directed. Relative expression was calculated using the $2^{-\Delta\Delta Ct}$ method with GAPDH (glyceraldehyde 3-phosphate dehydrogenase) as a reference gene. Final PCR products were resolved on a 2% agarose gel and imaged with an Azure Biosystems C-series gel documentation system (Dublin, CA) to confirm C3 expression in cell subsets.

### Statistical analysis

All data were evaluated using Prism six software (GraphPad, San Diego, CA). Statistical tests and post hoc analyses utilized are listed in each figure legend. Significance thresholds are indicated as follows $* = P < 0.05$, $** = P < 0.01$, $*** = P < 0.001$ for all data.

### Supplementary materials

Supporting data are available online.

## Acknowledgements

The authors thank Micaela Montgomery, Joan Kalnitsky, and Peter Saloupis for technical assistance. This project was supported by the following National Institutes of Health Grants: R01 EY021238 (DJJC), P30 EY021725 (University of Oklahoma Health Sciences Center), R01 EY021798 (DRS), P30 EY005722 (Duke University) and R01 EY024484 (VLP). Additional support was provided by unrestricted grants from Research to Prevent Blindness to the University of Oklahoma Health Sciences Center and Duke University. The content of this manuscript is solely the responsibility of the authors and does not necessarily represent the official views of the National Institutes of Health.

## Additional information

### Funding

| Funder | Grant reference number | Author |
|---|---|---|
| National Eye Institute | R01 EY021238 | Daniel JJ Carr |
| National Eye Institute | P30 EY021725 | Daniel JJ Carr |
| National Eye Institute | R01 EY021798 | Daniel R Saban |
| National Eye Institute | R01 EY024484 | Victor L Perez |
| National Eye Institute | P30 EY005722 | Daniel R Saban |

The funders had no role in study design, data collection and interpretation, or the decision to submit the work for publication.

## Author contributions
Derek J Royer, Conceptualization, Data curation, Software, Formal analysis, Validation, Investigation, Visualization, Methodology, Writing—original draft, Project administration, Writing—review and editing; Jose Echegaray-Mendez, Resources, Investigation; Liwen Lin, Rose Mathew, Investigation, Methodology; Grzegorz B Gmyrek, Investigation; Daniel R Saban, Victor L Perez, Supervision, Funding acquisition, Writing—review and editing; Daniel JJ Carr, Supervision, Funding acquisition, Investigation, Writing—review and editing

## Author ORCIDs
Derek J Royer (iD) https://orcid.org/0000-0002-8476-1784

## Ethics
Animal experimentation: This study was performed in strict accordance with the recommendations in the Guide for the Care and Use of Laboratory Animals of the National Institutes of Health. Research was performed in accordance with protocols approved by institutional care and use committees at the University of Oklahoma Health Sciences Center (Protocol number 160-014-NSI) and Duke University (Protocol numbers A061-18-03 and A034-18-01).

## Decision letter and Author response
Decision letter https://doi.org/10.7554/eLife.48378.023
Author response https://doi.org/10.7554/eLife.48378.024

# Additional files

## Supplementary files
• Transparent reporting form
DOI: https://doi.org/10.7554/eLife.48378.021

## Data availability
Data generated during this study are included in the manuscript.

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
