## [Decision Letter]

Thank you for submitting your article "Complement and CD4^+^ T cells drive context-specific corneal sensory neuropathy" for consideration by *eLife*. Your article has been reviewed by three peer reviewers, one of whom is a member of our Board of Reviewing Editors, and the evaluation has been overseen by Tadatsugu Taniguchi as the Senior Editor. The following individuals involved in review of your submission have agreed to reveal their identity: Mihaela Gadjeva (Reviewer #2); Susmit Suvas (Reviewer #3).

The reviewers have discussed the reviews with one another and the Reviewing Editor has drafted this decision to help you prepare a revised submission.

Summary:

In this interesting study, Derek et al. showed the involvement of complement C3 in causing corneal sensory neuropathy in mouse models of HSV keratitis, T-dependent allergic eye disease, and ocular graft-versus-host disease (GVHD). They find that complement C3 plays a key role in nerve damage during HSV-1 induced ocular keratitis, as well as in a mouse model of GVHD. They also find that T cells played a role in both disease conditions, and in particular CD4 T cells. Local depletion of C3 by cobra venom factor or the use of C3^-/-^ mice in these two models of chronic ocular inflammation prevented the loss of corneal sensation. The authors also showed that adoptive transfer of purified splenic CD4^+^ or CD8^+^ T cells from HSV-1 infected mice into TCRa^-/-^ mice documented the involvement of CD4 but not CD8 expressing T cells in causing the loss of corneal sensation after ocular HSV-1 infection. Together, these studies confirm the involvement of C3 and CD4 expressing T cells in causing the loss of corneal sensation in two major mouse models of chronic ocular inflammation. Therefore, these findings uncover immune mechanisms of nerve damage in several important eye inflammation contexts, and has interesting clinical implications for major conditions for patients that suffer from HSV1 and GVHD. All three reviewers agreed that this is a nice study and worthy of publication. However, there is a need to address some points and concerns raised below.

Essential revisions:

1) What is the relationship between C3 and T cells in HSV1 induced neuropathy? Figure 1F does not convincingly show equal (or decreased) influx of CD3+ T cells in HSV1 infected corneas from B6 and C3^-/-^ mice. There is a need to prove this by a more quantitative approach either by imaging or by flow cytometry.

2) The authors did not address potential non-specific effects of CVF while administering into infected corneas. Does it affect T cell survival or influx in HSV infected corneas?

3) Does CVF treatment decrease local T cell recruitment in the GVHD model? Figure 8E seems to indicate this, but can the authors quantify this by immunostaining or flow cytometry?

4) The baseline role for T cells in mediating HSV1 induced pain and neuropathy is not complete prior to adoptive transfers. In the HSV1 model, how do WT mice compare with TCRa^-/-^ mice for corneal sensation loss in the Cochet-Bonnet score? Is there less nerve damage in TCRa^-/-^ mice?

5) In Figure 3B, can the authors show the full time course of the corneal sensation threshold between WT vs. OT-II transfer data?

6) Generally, is the decrease in neuronal fibers associated with loss of neuronal cells in the TGs?

7) In view of recent work describing the importance of intracellular complement for T cell functionality, authors should interrogate the importance of intracellular complement synthesis and/or T cell activation in the models (Hansen et al., 2019; West et al., 2018). Indeed, authors show RT-PCR data from day 3 sorted cells from mice infected with HSV corneas, however, the dominant T-cell mediated effect seems to occur at later time points, suggestive a possible contribution of endogenous T-cell derived complement. Can the authors address this possibility?

8) If C3 is predominant factor in causing the loss of corneal sensation in these models, what is the possible underlying mechanism?

9) Can authors comment on the potential mechanisms of neuronal fiber retractions?

10) The use of CVF treatment offers a traditional approach to reduce complement activation. Given the developed therapies that target C5aR-signaling, authors should comment on the use of anti-C5a biologics for complement blockade in view of their discussion on potential therapies.

11) What possible factors from effector CD4 T cells are involved in activating C3 in GVHD and keratitis model?

12) While AED seems to be a contrasting model, the authors find lack of sensory phenotypes and nerve loss in this model (Figure 6). Therefore, we do not think this is a good model as comparison. The limitations of comparing this vs. the other models should be Discussed.

13) In Figure 1, it's interesting that while HSV-1 titers go up in the tear film and trigeminal ganglia (TG) at days 5 and 7 in C3^-/-^ mice compared to WT mice, HSV titers are not elevated in the C3^-/-^ cornea compared to WT mice. It's possible that the reason viral burdens are similar between WT and c3^-/-^ cornea is that a large number of nerve fibers and keratinocytes are destroyed that harbor the virus in WT mice. Could the authors address this point in the Discussion?

14) References: The authors can cite additional papers relevant to HSV-1 and complement C3: This includes the papers by Carroll and Knipe showing myeloid immune cell derived complement plays a role in HSV-1 host defense in peripheral tissues (Verschoor et al., 2001; Gadjeva et al., J Immunol. 2002 PMID: 12421924; Verschoor et al., J Immunol 2003 PMID:14607939) and papers showing that HSV-1 inhibits complement through glycoprotein C to evade host defense (Hook et al., J Virol 2006 PMID: 16571820; Lubinski et al., 2002)

---

## [Author Response]

Essential revisions:1) What is the relationship between C3 and T cells in HSV1 induced neuropathy? Figure 1F does not convincingly show equal (or decreased) influx of CD3+ T cells in HSV1 infected corneas from B6 and C3^-/-^ mice. There is a need to prove this by a more quantitative approach either by imaging or by flow cytometry.

We performed additional experiments to quantitatively assess T cell in the corneas of WT and C3^-/-^ mice by flow cytometry. These data show no difference in CD4 or CD8 T cell infiltration at day 7 post-infection(Figure 1G).

Briefly, the relationship between C3 and T cells is likely an indirect, coordinated process. Please see our working hypothesis for the underlying pathomechanism in critique #8 below.

2) The authors did not address potential non-specific effects of CVF while administering into infected corneas. Does it affect T cell survival or influx in HSV infected corneas?

Pharmacologic information about CVF is now included in the Discussion to clarify the mechanism of action and potential side effects. Briefly, CVF is generally nontoxic. While it has been used experimentally for several decades, there is little substantive evidence that it directly impacts the physiology of T cells. Please see Discussion paragraphs nine and ten for more information.

CVF treatment leads to a ~50% reduction in cornea-infiltrating CD4 T cells compared to the vehicle control group (Figure 5G). This is consistent with the observation that CVF treatment also reduces corneal inflammation/edema (Figure 5D).

CVF treatment had no discernable impact on T cell expansion in the draining lymph nodes or numbers in peripheral circulation (Figure 5—figure supplement 2). Taken together, this suggests that CVF does not alter T cell survival.

3) Does CVF treatment decrease local T cell recruitment in the GVHD model? Figure 8E seems to indicate this, but can the authors quantify this by immunostaining or flow cytometry?

T cell numbers and nerve densities were quantified from confocal images of cornea flat mounts using Imaris software (AED and GVHD models). These data are now included in Figures 6, 7, and 8.

Briefly, no differences were found in the numbers of CD3^+^ cells in the central corneas of CVF vs. PBS treated mice in the GHVD model at the experiment endpoint (Figure 8H). Given the protracted treatment regimen, this suggests that CVF likely has no impact on T cell survival (relevant to critique # 2 above).

Our data does not exclude the possibility that CVF treatment may delay T cell influx. However, the Perez laboratory is currently investigating T cell response dynamics in the progression of ocular GVHD.

4) The baseline role for T cells in mediating HSV1 induced pain and neuropathy is not complete prior to adoptive transfers. In the HSV1 model, how do WT mice compare with TCRa^-/-^ mice for corneal sensation loss in the Cochet-Bonnet score? Is there less nerve damage in TCRa^-/-^ mice?

Cochet-Bonnet scores for corneal sensation are highly reproducible direct measurements (data are not relative/normalized). As shown in the original manuscript, sensation loss was not observed in TCRα^-/-^ mice without addition of exogenous T cells.

Corneal nerve imaging was not performed on corneas from TCRα^-/-^ mice, but the sensation data underscore that HSV-associated sensation loss is T cell-dependent.

The initial time (day post-infection) at which sensation is ‘significantly’ reduced relative to the corresponding control lags by one day in the adoptive transfer model compared to WT mice. This corresponds with the relative lymphopenic status of the TCRα^-/-^ mice following adoptive transfer compared to WT mice, which have more than 10 times as many T cells in the cornea-draining lymph nodes at day 8 post-infection (compare Figure 2B and Figure 2—figure supplement 1A). However, the tempo of sensation loss is consistent between models once instigated by HSV-1 infection.

From a gross pathology standpoint, corneas from TCRα^-/-^ control mice generally remained transparent following HSV-1 infection. However, corneal opacification was observed in HSV-1-infected TCRα^-/-^ mice reconstituted with exogenous T cells (data not shown). These observations parallel those from the Hendricks lab showing a reduction in corneal opacification when mice are depleted of CD4 T cells (Yun et al., 2014).

5) In Figure 3B, can the authors show the full time course of the corneal sensation threshold between WT vs. OT-II transfer data?

Baseline sensation and control data are now included in Figure 3B and Figure 3—figure supplement 2.

These experiments were performed to determine the necessity of antigen-specific T cells in driving sensation loss after the tempo of corneal sensation loss had been established (Figure 1B and Figure 3A). Accordingly, corneal sensation data were only measured at baseline and day 7 post-infection for the experiments involving OT-II mice and cells.

6) Generally, is the decrease in neuronal fibers associated with loss of neuronal cells in the TGs?

The cell bodies of the corneal nerves are housed within the trigeminal ganglion. Accordingly, addressing these related issues would require meticulous nerve tracing studies well beyond the scope of the current manuscript.

The current investigation specifically addresses immune-mediated mechanisms of corneal sensation loss using murine models of clinically-relevant ocular surface inflammatory diseases. While certainly important, the individual fates of damaged corneal sensory nerves, were not explored.

The revised Discussion includes a brief synopsis of the relevant literature:

“…the individual fates of damaged corneal nerves were not explored herein. Lessons from peripheral nerve injury in other settings indicate that some damaged nerves undergo apoptosis, yet effectively promoting functional regeneration of surviving neurons remains a major clinical hurtle… Likewise, promoting functional regeneration of damaged corneal nerves is important for restoration of immune privilege and ocular surface health.”

7) In view of recent work describing the importance of intracellular complement for T cell functionality, authors should interrogate the importance of intracellular complement synthesis and/or T cell activation in the models (Hansen et al., 2019; West et al., 2018). Indeed, authors show RT-PCR data from day 3 sorted cells from mice infected with HSV corneas, however, the dominant T-cell mediated effect seems to occur at later time points, suggestive a possible contribution of endogenous T-cell derived complement. Can the authors address this possibility?

Intracellular complement function is intriguing and certainly noteworthy. However, this area remains somewhat controversial within the complement field (discussed extensively at the 2018 International Complement Workshop in Santa Fe, NM).

The revised manuscript Discussion includes citations for the following landmark studies, and the pertinent findings are outlined below:

a) Claudia Kemper’s lab indicates that murine CD4 T cells lacking C3 have reduced Th1 responses (IFNγ) following in vitroexpansion and re-stimulation (Sup. Figure 4I in Liszewski et al., 2013).

– This paper’s primary finding is that C3a may have a critical survival function on *human* T cells, but this has not been recapitulated in mice.

– Similarly,Th1 responses are reduced in C3-deficient human T cells following in vitro stimulation (PMID: 24321396).

b) Peter Garred’s lab utilized human donor cells and found that C3 product/receptor expression were *downregulated* in T cells activated in vitro. They conclude that “endogenous[ly] expressed C3 is *not* vital for T cell turnover” (Hansen et al., 2019).

c) John Atkinson’s lab has shown that human CD4 T cells acquire C3 from the extracellular environment, but exogenous C3 did not alter IFNγ production upon activation in vitro (Elvington et al., 2017).

Importantly, the studies cited above investigate Th1 responses in human T cells with anti-CD3 and anti-CD46 stimulation in vitro. There are species-specific differences in complement biology between humans and mice:

a) While CD46 is required for Th1 polarization/cytokine production in human T cells, mice do not express CD46.

b) Instead, data from the Medof lab indicate that anaphylatoxin receptors are important for Th1 cytokine production in mice (Strainic et al., 2008).

In light of the studies listed above, we addressed several specific questions relative to C3 and T cell function using the HSV-1 ocular infection model as a way to prime T cells in vivo.

a) Our data show no difference in IFNγ production in T cells from C3^-/-^ and WT mice following in vivo priming by way of ocular HSV-1 infection (Figure 2C). This was determined using a PMA/ionomycin-based re-stimulation assay comparable to that used in the studies cited above.

b) Our data show that endogenous C3 production is not required for T cell functionality in mice. This is based on the ability of sorted C3^-/-^ T cells to mediate corneal sensation loss upon transfer into TCRα^-/-^ mice (Figure 2D).

c) The cornea contains high concentrations of C3 protein (Figure 5B) prior to T cell infiltration. Accordingly, there is little evidence that putative C3 production by murine T cells specifically tips the balance in favor of pathology.

d) It remains possible that anaphylatoxin receptor signaling from locally produced C3a (and potentially C5a generated downstream) modulates T cell effector function upon extravasation into the inflammatory milieu. However, anaphylatoxin receptor signaling was not investigated herein.

Finally, C3 breakdown products may lower the threshold for T cell activation/effector function through costimulatory pathways (e.g. CD46, C3aR1) akin to what has been shown for B cell responses through CD21 (PMID: 18328742, 30305631, 11418645, 25210145).

a) The clinical phenotype in the C3^-/-^ human population, albeit rare, involves high susceptibility to bacterial infections and immune complex-mediated diseases. However, published evidence suggesting that C3^-/-^ patients are more susceptible to viral infections is conspicuously absent (PMID: 16499568, 31192812, 18802120).

b) If a profound Th1 deficit existed in C3^-/-^ patients, their clinical phenotype would almost certainly include maladaptive responses to viral infections with increased morbidity.

c) Importantly, C3b and C4b are both ligands for CD46, thus C4b may compensate for the absence of C3b to facilitate Th1 induction in C3-deficient patients in vivo.

8) If C3 is predominant factor in causing the loss of corneal sensation in these models, what is the possible underlying mechanism?

The revised discussion now outlines a working hypothesis for the possible mechanism of corneal sensation loss (Discussion paragraph seven). This framework is a context-specific process involving C3 and CD4 T cells and is supported by established findings:

a) Corneal inflammation favors local activation of the complement pathway.

b) Sensory nerve terminals are damaged by sublytic MAC formation downstream of C3 activation (PMID: 21850539).

c) Cytokines produced by CD4 T cells (Th1) activate phagocytes (professional and nonprofessional) to consume damaged sensory nerve fibers.

d) Local complement activation may also augment Th1 cytokine secretion upon extravasation into the cornea via anaphlatoxin receptor signaling (PMID: 18328742).

The relevant complement pathways (classical, alternative, lectin, MAC-involvement), cellular targets, and the impact of anaphylatoxin receptor signaling are all topics of future investigation. By addressing the ‘heart’ of the complement pathway (C3), this study opens the door to future investigations to further elucidate the relevant pathomechanisms and corresponding therapeutic targets.

9) Can authors comment on the potential mechanisms of neuronal fiber retractions?

The nerve-intrinsic mechanism underlying sensory fiber retraction has not been clearly established for any corneal pathology to date. In line with the mechanism outlined above (critique #8), we hypothesize that membrane attack complex deposition on sensory fibers leads to damage and clearance. This is described in the revised manuscript:

“[An] imbalance in ‘complement proteostasis’ may favor deposition of complement fragments and sublytic MAC on corneal sensory nerve fibers (Tegla et al., 2011; Triantafilou et al., 2013). Notwithstanding, the nerve-intrinsic molecular mechanisms responsible for trigeminal sensory fiber retraction, neuronal death, or axonal regeneration in the cornea are largely unknown Stepp et al., 2017).”

Also, please see response to critique #6.

10) The use of CVF treatment offers a traditional approach to reduce complement activation. Given the developed therapies that target C5aR-signaling, authors should comment on the use of anti-C5a biologics for complement blockade in view of their discussion on potential therapies.

CVF was used merely as an experimental proof of concept. Subsequent studies are needed to investigate each complement initiation pathway, other effectors, and anaphylatoxin receptors.

Please see the “complement drug development pipeline” graphic featured in Ricklin et al., 2018.

There are many complement-targeted therapeutics in pre-clinical development or clinical trials (see the “complement drug development pipeline” graphic featured in Ricklin et al., 2018). The anti-C5 antibody Eculizumab is perhaps the most successful of the approved complement therapeutics, but its effect on C5aR signaling is indirect. In light of the numerous complement targets and concordant drugs in preclinical development, a particular focus on C5aR-signaling is hard to justify in the discussion, as our paper does not address C5 directly.

11) What possible factors from effector CD4 T cells are involved in activating C3 in GVHD and keratitis model?

Please see response to critique #8 above.

CD4 T cells are not required for C3 activation in the cornea. Our hypothesis is that T cell cytokines activate phagocytes to clear complement-tagged sensory nerve fibers.

This topic has been addressed in the revised Discussion:

“AED is driven primarily by a combined Th2/Th17 CD4 T cell response that differs from the Th1 bias observed in HSV-1 keratitis and ocular GHVD. This suggests that Th1 cytokines such as IFNγ, IL-2, and lymphotoxin-α may be important in coordinating corneal sensation loss.”

12) While AED seems to be a contrasting model, the authors find lack of sensory phenotypes and nerve loss in this model (Figure 6). Therefore, we do not think this is a good model as comparison. The limitations of comparing this vs. the other models should be Discussed.

Differences in CD4 T cell polarization among the models are now clearly highlighted in the Discussion (please see response 11).

To further address this point, we have added the following to the Discussion:

“The lack of sensory phenotype in the AED model was unanticipated, given that patients with severe allergy have been shown to have changes in nerve morphology… This contrasting model underscores the context-dependency of sensation loss during corneal inflammation.”

13) In Figure 1, it's interesting that while HSV-1 titers go up in the tear film and trigeminal ganglia (TG) at days 5 and 7 in C3^-/-^ mice compared to WT mice, HSV titers are not elevated in the C3^-/-^ cornea compared to WT mice. It's possible that the reason viral burdens are similar between WT and c3^-/-^ cornea is that a large number of nerve fibers and keratinocytes are destroyed that harbor the virus in WT mice. Could the authors address this point in the Discussion?

This topic has been addressed in the revised discussion in parallel with a commentary on the impact of HSV-1 glycoprotein C (Discussion section paragraph four).

Briefly, it is likely that the potent antiviral type-1 interferon response protects the cornea against excessive HSV-1 replication even in the absence of C3. Accordingly, the maintenance of corneal innervation in C3^-/-^ mice may enhance shedding of virus produced within the TG.

14) References: The authors can cite additional papers relevant to HSV-1 and complement C3: This includes the papers by Carroll and Knipe showing myeloid immune cell derived complement plays a role in HSV-1 host defense in peripheral tissues (Verschoor et al.,, 2001; Gadjeva et al., J Immunol. 2002 PMID: 12421924; Verschoor et al., J Immunol 2003 PMID:14607939) and papers showing that HSV-1 inhibits complement through glycoprotein C to evade host defense (Hook et al., J Virol 2006 PMID: 16571820; Lubinski et al., 2002)

In our model, corneal nerve damage occurs prior to generation of broad humoral immunity to HSV-1. The Carroll and Knipe papers cited above show the impact of bone marrow-derived complement on *induction* of humoral immunity to HSV-1, but they do not address host defense with respect to viral pathogenesis or tissue damage. The papers are solid, but their relationship to the current investigation is debatable.

However, the revised manuscript cites the Verschoor et al. paper (PMID: 11509581) as evidence of macrophage-derived C3 production and a 1999 *PNAS* paper from Knipe and Carroll corroborating normal proliferation of WT and C3^-/-^ T cells primed by vaccination with a replication defective virus based on [^3^H]thymidine uptake in response to viral antigen re-stimulation (PMID: 10535987).

The topic of glycoprotein C (gC) and complement evasion are now addressed in the Discussion (paragraph four).